# Bayesian Optimisation with Unknown Hyperparameters: Regret Bounds Logarithmically Closer to Optimal

**Juliusz Ziomek**[†,★]**, Masaki Adachi**[†,‡]**, Michael A. Osborne**[†]**,**

[†]Machine Learning Research Group, University of Oxford

[‡] Toyota Motor Corporation

[★] Corresponding Author

`{juliusz, masaki, mosb}@robots.ox.ac.uk`

## Abstract

Bayesian Optimization (BO) is widely used for optimising black-box functions but requires us to specify the length scale hyperparameter, which defines the smoothness of the functions the optimizer will consider. Most current BO algorithms choose this hyperparameter by maximizing the marginal likelihood of the observed data, albeit risking misspecification if the objective function is less smooth in regions we have not yet explored. The only prior solution addressing this problem with theoretical guarantees was A-GP-UCB, proposed by Berkenkamp et al. (2019). This algorithm progressively decreases the length scale, expanding the class of functions considered by the optimizer. However, A-GP-UCB lacks a stopping mechanism, leading to over-exploration and slow convergence. To overcome this, we introduce Length scale Balancing (LB)—a novel approach, aggregating multiple base surrogate models with varying length scales. LB intermittently adds smaller length scale candidate values while retaining longer scales, balancing exploration and exploitation. We formally derive a cumulative regret bound of LB and compare it with the regret of an oracle BO algorithm using the optimal length scale. Denoting the factor by which the regret bound of A-GP-UCB was away from oracle as $g(T)$, we show that LB is only $\log g(T)$ away from oracle regret. We also empirically evaluate our algorithm on synthetic and real-world benchmarks and show it outperforms A-GP-UCB, maximum likelihood estimation and MCMC.

## 1 Introduction

Bayesian Optimisation (BO) [16] has proven to be an efficient solution for black-box optimisation problems, finding applications across science, engineering and machine learning [12, 18, 23]. As a model-based optimisation technique, BO constructs a surrogate model of the black box function, which is typically a Gaussian Process (GP) [40]. However, to construct this surrogate, we need to specify our expectations about the smoothness of the black-box function. In the case of GP, the choice of smoothness is reflected in the selection of an appropriate length scale value for the kernel function. Selecting smaller length scales allows us to model less smooth functions and, as such, expands the class of all possible black-box functions the optimiser will consider. At the same time, it makes the convergence of the algorithm slower, due to an increase in the number of possible 'candidate' functions, the algorithm has to explore the space much more. As such, we wish to consider the smallest possible class of functions that still contains the black-box function we wish to solve. This translates to selecting some optimal length scale value, which is neither too short nor too long.

38th Conference on Neural Information Processing Systems (NeurIPS 2024).

Appropriate selection of the length scale parameter can be challenging. Typical practice is to fit the length scale value by maximum likelihood estimation (MLE) on the observed data we have collected thus far. However, it is entirely possible that the function changes less smoothly in the regions we have not explored yet, as shown in Figure 1. As such, we cannot guarantee that maximising the likelihood of the limited, observed data will find a length scale value such that the black-box function will lie in the space of considered functions. A previously proposed algorithm called A-GP-UCB [6] approached this issue by progressively decreasing the length scale value, and as such increasing the class of functions considered by the optimiser. As a consequence, at some point, it must contain the black-box function we are trying to optimise. However, the algorithm has no mechanism for stopping and as such the length scale value will decrease indefinitely, inexorably expanding the class of considered functions. This causes over-exploration, making the convergence much slower compared to an optimiser that knows the optimal length scale value.

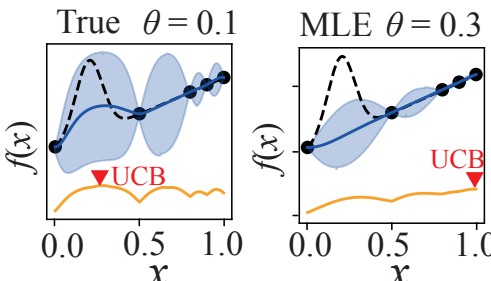

Figure 1: An objective function (proposed by [6]) that illustrates the importance of length scales to BO. The blue line shows a GP fit with shaded regions representing one standard deviation. The length scale value was set to the optimal value on the left and was selected by MLE on the right, based on five points represented by dots. While the optimiser with the MLE of length scale persistently selects a suboptimal value of $x = 1$, the optimiser with the optimal length scale can spot the hidden peak leading to finding the maximum at $x^* = 0.3$.

A-GP-UCB is suboptimal because it never returns to previously trialled, longer length scales. Observe that if trying a shorter length scale value does not improve function discovery, opting for a longer scale is a safer choice, preventing excessive exploration. To build an algorithm following this intuition, we could have a number of base optimisers, each utilising a different length scale value, and aggregate them into a single 'master' optimiser. By knowing how explorative each of the base optimisers is, the 'master' optimiser could select the most suitable one at each iteration, so as to balance exploration and exploitation. Within the literature of multi-armed bandit problems, a number of rules for aggregating base algorithms have been proposed [1, 3, 33], however, the performance of those 'master' algorithms worsens with the number of base algorithms. This prohibits us from directly applying 'master' algorithms to the unknown hyperparameter problem, as the length scale is a continuous parameter and has infinitely many possible values.

Within this work, we extend one such aggregation scheme, called regret-balancing, to handle infinitely many learners, so that it could tackle the problem of BO with unknown hyperparameters. We propose an algorithm called Length scale Balancing GP-UCB (LB-GP-UCB), which aggregates a number of base optimisers with different length scale values and gradually introduces new base optimisers, equipped with smaller length scales. Instead of permanently decreasing the length scale value, as done by A-GP-UCB, LB-GP-UCB occasionally introduces new base learners with smaller length scale values, while still maintaining base learners with longer ones. As such, if one of the longer length scales is optimal, we will be able to recover performance close to the one of the oracle optimiser utilising that optimal length scale. Denoting the factor by which the regret bound of A-GP-UCB was away from oracle as $g(T)$, we show that LB-GP-UCB is only $\log g(T)$ away from oracle regret. We also conduct empirical evaluation and show LB-GP-UCB obtains improved regret results on a mix of synthetic and real-world benchmarks com-

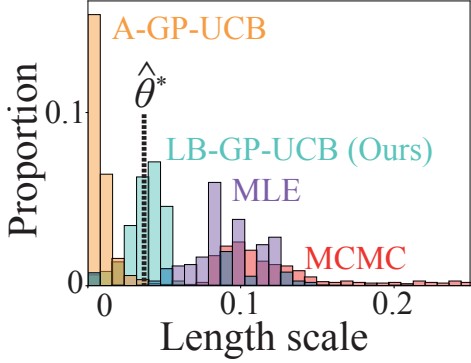

Figure 2: Histogram showing how often (as a proportion of iterations) each algorithm selected a given length scale value while optimising the Michalewicz function over ten seeds. $\hat{\theta}^\star$ corresponds to an estimate of optimal length scale value. See §5 for details.

pared to A-GP-UCB, MLE and MCMC. We show the histogram of length scale values selected by each method on one of the benchmark problems in Figure 2. $\hat{\theta}^\star$ represents an estimate of optimal length scale value on this problem (see §5 for details). We can see that the length scale values selected by LB-GP-UCB are close to the estimated optimal value, whereas MLE and A-GP-UCB miss this value by respectively over and under-estimating, matching our predictions. We summarise our contributions below.

- We propose LB-GP-UCB and show that compared to A-GP-UCB its regret bound is logarithmically closer to the bound of an oracle optimiser knowing the optimal length scale.
- We extend our algorithm to also handle the case of unknown output scale (function norm) alongside the length scale
- We show the empirical superiority of our algorithm compared to MLE, MCMC and A-GP-UCB on a mix of synthetic and real-world problems, and conduct an ablation study showing increased robustness of our method.

## 2    Problem Statement and Preliminaries

We consider the problem of maximising an unknown black-box function $f : \mathcal{X} \to \mathbb{R}$ on some compact set $\mathcal{X} \subset \mathbb{R}^d$. At each time step $t$, we are allowed to query a function at a selected point $\boldsymbol{x}_t \in \mathcal{X}$ and observe noisy feedback $y_t = f(\boldsymbol{x}_t) + \epsilon_t$, where $\epsilon_t \sim \text{SubGauss}(\sigma_N^2)$. We wish to find the optimum $\boldsymbol{x}^\star = \max_{\boldsymbol{x} \in \mathcal{X}} f(\boldsymbol{x})$. We define the instantaneous regret to be $r_t = f(\boldsymbol{x}^\star) - f(\boldsymbol{x}_t)$ and cumulative regret as $R_T = \sum_{t=1}^{T} r_t$, and we wish to minimise it. We assume we are given some kernel function $k^\theta(\boldsymbol{x}, \boldsymbol{x}') = k(\frac{\boldsymbol{x}}{\theta}, \frac{\boldsymbol{x}'}{\theta})$ parametrised by a length scale value $\theta \in \mathbb{R}^+$ and we denote its associated Reproducing Kernel Hilbert Space (RKHS) as $\mathcal{H}(k^\theta)$. We assume that at least for certain values of $\theta$, the black-box function $f$ belongs to this RKHS, i.e. $\exists_{\theta \in \mathbb{R}^+} f \in \mathcal{H}(k^\theta)$. Concretely, we will consider two popular types of kernels: RBF and $\nu$-Matérn, defined below for completeness:

$$k_{\text{RBF}}^\theta(\boldsymbol{x}, \boldsymbol{x}') = \exp\left(-\frac{\|\boldsymbol{x} - \boldsymbol{x}'\|_2^2}{\theta^2}\right)$$

$$k_{\nu\text{-Matérn}}^\theta(\boldsymbol{x}, \boldsymbol{x}') = \frac{2^{1-\nu}}{\Gamma(\nu)}\left(\sqrt{2\nu}\frac{\|\boldsymbol{x} - \boldsymbol{x}'\|}{\theta}\right)^\nu K_\nu\left(\sqrt{2\nu}\frac{\|\boldsymbol{x} - \boldsymbol{x}'\|}{\theta}\right),$$

where $\Gamma(\cdot)$ is the Gamma function and $K_\nu(\cdot)$ is the modified Bessel function of the second kind of order $\nu$. Without the loss of generality, we assume $k^\theta(\cdot, \cdot) \leq 1$ for all $\theta \in \mathbb{R}^+$. In the rest of the paper, if we do not specify the type of the kernel, it means the result is applicable to both types of kernels. If we fit GP model with a kernel $k^\theta(\boldsymbol{x}, \boldsymbol{x}')$ to the data so far, $\mathcal{D}_{t-1} = \{(\boldsymbol{x}_\tau, y_\tau)\}_{\tau=1}^{t-1}$, we obtain the following mean $\mu_{t-1}^\theta(\boldsymbol{x})$ and variance $(\sigma_{t-1}^\theta)^2(\boldsymbol{x})$ functions:

$$\mu_{t-1}^\theta(\boldsymbol{x}) = \boldsymbol{k}_{t-1}^\theta(\boldsymbol{x})^T(\mathbf{K}_{t-1}^\theta + \sigma_N^2\mathbf{I})^{-1}\boldsymbol{y}_{t-1}$$

$$(\sigma_{t-1}^\theta)^2(\boldsymbol{x}) = k^\theta(\boldsymbol{x}, \boldsymbol{x}) - \boldsymbol{k}_{t-1}^\theta(\boldsymbol{x})^T(\mathbf{K}_{t-1}^\theta + \sigma_N^2\mathbf{I})^{-1}\boldsymbol{k}_{t-1}^\theta(\boldsymbol{x}),$$

where $\boldsymbol{y}_{t-1} \in \mathbb{R}^{t-1}$ with elements $(\boldsymbol{y})_i = y_i$, $\boldsymbol{k}^\theta(\boldsymbol{x}) \in \mathbb{R}^{t-1}$ with elements $\boldsymbol{k}^\theta(\boldsymbol{x})_i = k^\theta(\boldsymbol{x}, \boldsymbol{x}_i)$ and similarly $\mathbf{K}_{t-1}^\theta \in \mathbb{R}^{t-1 \times t-1}$ with entries $(\mathbf{K}_{t-1}^\theta)_{i,j} = k^\theta(\boldsymbol{x}_i, \boldsymbol{x}_j)$, and $\sigma_N^2$ is the regulariser factor [11] with identity matrix $\mathbf{I}$. If we were to use any length scale value such that $f \in \mathcal{H}(k^\theta)$, then we can obtain certain guarantees about the predictions made by the GP model, as stated next.

**Theorem 2.1** (Theorem 2 of [11]). *Let $f \in \mathcal{H}(k^\theta)$, such that $\|f\|_{k^\theta} \leq B$ and set $\beta_t^{\theta,B} = B + \sigma_N\sqrt{2(\mathcal{I}_{t-1}(k^\theta) + 1 + \ln(1/\delta_A))}$, where $\mathcal{I}_T(k^\theta)$ is an upper bound $\frac{1}{2}\log|I + \sigma_N^{-2}\mathbf{K}_T^{\theta^\star}| \leq \mathcal{I}_T(k^\theta)$, which depends on the kernel and length scale choice. Then, with probability at least $1 - \delta_A$, for all $\boldsymbol{x} \in \mathcal{X}$ and $t = 1, \ldots, T$:*

$$\left|f(\boldsymbol{x}) - \mu_{t-1}^\theta(\boldsymbol{x})\right| \leq \beta_t^{\theta,B}\sigma_{t-1}^\theta(\boldsymbol{x}).$$

Note that the Theorem 2.1 relies on the quantity $\mathcal{I}_T(k^\theta)$, also called maximum information gain (MIG). The next proposition, proven in Appendix A, provides bounds on $\mathcal{I}_T(k^\theta)$ for RBF and $\nu$-Matérn kernel and shows the explicit dependence on length scale hyperparameter $\theta$.

**Proposition 2.2.** *We have that $\mathcal{I}_T(k^\theta) \le \mathcal{O}\left(\gamma_T(k^\theta)\right)$:*

- *For an RBF kernel $\gamma_T(k^\theta) = \frac{1}{\theta^d}\log(T)^{d+1}$*

- *For $\nu$-Matérn kernel $\gamma_T(k^\theta) = \frac{1}{\theta^d}T^{\frac{d(d+1)}{2\nu+d(d+1)}}\log(T)^{\frac{2\nu}{2\nu+d}}$*

Based on the GP model, a typical BO algorithm constructs an acquisition function, which tells us how 'promising' a given point is to try next. We will focus on the commonly used Upper Confidence Bound (UCB), defined as $\text{UCB}_t^{\theta,B}(\boldsymbol{x}) = \mu_{t-1}^\theta(\boldsymbol{x}) + \beta_t^{\theta,B}\sigma_{t-1}^\theta(\boldsymbol{x})$. The GP-UCB algorithm [36, 11] fits a GP model and utilises the UCB criterion to select new points to query. Such an algorithm admits a high-probability regret bound as stated by the next Theorem.

**Theorem 2.3** (Theorem 2 in [11]). *Let us run a GP-UCB utilising a GP with a kernel $k^\theta$ and the exploration bonus of $\beta_t^{\theta,B} = B + \sigma_N\sqrt{2(\mathcal{I}_T(k^\theta)) + 1 + \ln(1/\delta_A))}$ on a black-box function $f \in \mathcal{H}(k^\theta)$ such that $\|f\|_{k^\theta} \le B$. Then, with probability at least $1-\delta_A$, it admits the following bound on its cumulative regret $R_T \le \mathcal{O}\left(R^{\theta,B}(T)\right)$, where $R^{\theta,B}(T) = \sqrt{T}\left(B\sqrt{\gamma_T(k^\theta)} + \gamma_T(k^\theta)\right)$.*

In our notation, note the distinction between the regret of an algorithm $R_T$ and the scaling of its bound $R^{\theta,B}(T)$. As the maximum regret we can possibly suffer at any time step, while optimising a function with property $\|f\|_{k^\theta} \le B$, is bounded as $r_t \le 2B$ [1], we are going to assume the bound obeys the property $R^{\theta,B}(t+1) - R^{\theta,B}(t) \le 2B$ for all $t = 1, \ldots, T-1$, as otherwise the bound can be trivially improved.

In order for the bound of Theorem 2.1 to hold, we need to know the length scale $\theta$ and an upper bound on the RKHS norm $B$ of the black-box function for the given kernel $k^\theta$. Inspecting the regret bound together with Proposition 2.2, we see that selecting the smallest $B$ (i.e. the tightest bound) and the longest length scale $\theta$ results in the smallest $R^{\theta,B}(T)$. Note that the same function $f(\cdot)$, can have different RKHS norms under kernels with different length scale values. As such, to obtain the optimal scaling of the regret bound, one needs to jointly optimise for $\theta$ and $B$. The optimal hyperparameters are thus $\theta^\star, B^\star = \arg\min_{\theta,B\in\mathbb{R}^+} R^{\theta,B}(T)$ such that $\|f\|_{k^{\theta^\star}} \le B^\star$. We assume we are given some initial $\theta_0 \ge \theta^\star$ and $B_0 \le B^\star$. As explained in the introduction, in practice, those initial values could be found by maximising the marginal likelihood for a small number of initial data points. We now notice one interesting property. In the case of RBF and $\nu$-Matérn kernels, if we change the length scale value from $\theta_0$ to $\theta$ and the norm bound from $B_0$ to $B$, we get that the regret bound with those new hyperparameters scales as follows:

$$R^{\theta,B}(T) = \sqrt{T}\left(\left(\frac{B}{B_0}\right)\left(\frac{\theta_0}{\theta}\right)^{d/2} B_0\sqrt{\gamma_T(k^{\theta_0})} + \left(\frac{\theta_0}{\theta}\right)^d \gamma_T(k^{\theta_0})\right).$$

Since $\gamma_T(k^\theta)$ is increasing in $T$, for large enough $T$ we have $B < \sqrt{\gamma_T(k^\theta)}$ and any $B < \left(\frac{\theta_0}{\theta}\right)^{d/2}B_0$ does not affect the bound's order dependence. As such, whenever we decrease lengthscale to $\theta$, we can increase norm bound by $\left(\frac{\theta_0}{\theta}\right)^{d/2}$ essentially "for free". As such, we are going to use $\theta$-dependent norms in the form of $B(\theta, N) = \left(\frac{\theta_0}{\theta}\right)^{d/2}N$, where $N$ is the norm bound under $\theta_0$ and becomes the new hyperparameter we wish to select, instead of $B$. The optimal values of hyperparameters under this new parameterization are thus $\theta^\star, N^\star = \arg\min R^{\theta,B(\theta,N)}(T)$ subject to $\|f\|_{k^{\theta^\star}} \le B(\theta^\star, N^\star)$. Notice that $\min_{\theta\in(0,\theta_0]} B(\theta, N^\star) = N^\star$ and as such it does not make sense to try values of $N$ smaller than $B_0$. Using this new parameterization brings an important benefit, as stated next.

**Lemma 2.1** (Consequence of Lemma 4 in [9]). *In case of RBF and Matérn kernels, for any $\theta < \theta^\star$ and $N > N^\star$, we have that $\|f\|_{k^\theta} \le B(\theta, N)$.*

We will thus refer to any pair $(\theta, N)$, such that $\theta \le \theta^\star$ and $N \ge N^\star$, as *well-specified* hyperparameters, as the GP-UCB admits a provable regret bound when they are used (albeit that bound might not be optimal). For simplicity, we are now going to assume that $N^\star$ is known and proceed with solving the problem of only one unknown hyperparameter $\theta^\star$. We will thus be writing $\beta_t^\theta = \beta_t^{\theta,B(\theta,N^\star)}$, $\text{UCB}_t^\theta(\cdot) = \text{UCB}_t^{\theta,B(\theta,N^\star)}(\cdot)$ and $R^\theta(\cdot) = R^{\theta,B(\theta,N^\star)}(\cdot)$. However, we would like to emphasise that the algorithm we will propose throughout this paper can be extended to the case when $B^\star$ is also an unknown hyperparameter, which we do in Appendix F.

---

[1]This is because $f(\boldsymbol{x}^\star) - f(\boldsymbol{x}_t) \le 2\|f\|_\infty \le 2\|f\|_{k^\theta} \le 2B$.

# 3 Length scale Balancing

Aggregation schemes describe a set of rules that a master algorithm should follow while coordinating a number of base algorithms. One such scheme is regret-balancing with elimination [33]. This scheme assumes each of the base algorithms comes with a *suspected regret bound*, which is a high-probability bound on its regret that holds if the algorithm is well-specified for the given problem, but might not hold if the learner is misspecified. The scheme always selects the base algorithm that currently has the smallest cumulative regret according to its suspected bound. This ensures that the regret of the master algorithm will not be too far from the regret of the best well-specified candidate. It also removes base algorithms that underperform, compared to others, by more than their suspected bound, as this means their bounds do not hold and, with a high probability, are misspecified.

Our idea is to use regret balancing with elimination while having each base algorithm be a GP-UCB algorithm with a different value of the length scale hyperparameter. Let us now discuss how to identify candidates for the length scale values. We propose the usage of a candidate-suggesting function $q(\cdot) : \mathbb{N} \to \mathbb{R}^+$, such that the $i$th candidate length scale value to consider is given by $q(i)$.

**Definition 3.1.** Let us define the length scale candidate-suggesting function $q(\cdot) : \mathbb{N} \to \mathbb{R}^+$ as a mapping for each $i \in \mathbb{N}$ of form:

$$q(i) = \theta_0 e^{-i/d}.$$

We want to ensure that one of the candidates we will eventually introduce will be close to $\theta^\star$. Let us denote $\hat{\theta} = \arg\max_{i \in \mathbb{N}\,;\,q(i) \le \theta^\star} q(i)$ to be the largest length scale suggested by our candidate-suggesting function that is still smaller than $\theta^\star$. Observe that $\|f\|_{k^{\hat{\theta}}} \le B(\hat{\theta}, N^\star)$ and thus $\hat{\theta}$ is the largest well-specified length scale value among suggested candidates. As such, the regret bound of the best base learner is $R^{\hat{\theta}}(T)$ and we hope that the regret bound of a master algorithm aggregating this learner with others will be close to $R^{\hat{\theta}}(T)$. Comparing with the regret bound of the GP-UCB algorithm utilising the true optimal length scale value $R^{\theta^\star}(T)$, we get the result stated by the following Lemma 3.1, proven in Appendix C

**Lemma 3.1.** *In the case of both RBF and $\nu$-Matérn kernel, we have that:*

$$\frac{R^{\hat{\theta}}(T)}{R^{\theta^\star}(T)} = \mathcal{O}(1).$$

This Lemma shows that the regret bound of the best of our base algorithms is only a constant factor away from the bound of the algorithm using the optimal length scale value. However, as for any $i \in \mathbb{N}$, we have $q(i) > 0$, and the candidate-suggesting function $q(\cdot)$ introduces infinitely many candidates. As we can only aggregate a finite number of base algorithms, we thus propose to gradually introduce new optimisers equipped with new candidate length scale values. Observe that if we stopped our quantisation at some lower bound $\theta_L$, then we would create a maximum of $q^{-1}(\theta_L)$ candidates, that is, $d \ln(\frac{\theta_0}{\theta_L})$. However, this would require us to know a sure lower bound on the optimal length scale value. Since we do not have this knowledge, we could employ a mechanism similar to A-GP-UCB, where we progressively decrease the *suspected lower bound value* $\theta_L(t) = \frac{\theta_0}{g(t)}$, based on some growth function $g(t)$. Observe that since $\ln\left(\frac{\theta_0}{\theta_L(t)}\right) = \ln(g(t))$, the number of candidate values grows only logarithmically with the growth function $g(t)$. Same as for A-GP-UCB, this growth function needs to be specified by the user, and we describe how this choice can be made in §5. However, we would like to emphasise that, unlike A-GP-UCB which simply sets its length scale value to $\theta_L(t)$, we instead introduce new learners with shorter length scale values, while still keeping the old learners with longer values. This strategy is thus more robust to the choice of the growth function, which is reflected in better scaling of the regret bound we derive later. In Algorithm 1, we present LB-GP-UCB, an algorithm employing this mechanism. We now briefly explain the logic behind its operations.

The algorithm starts in line 1 by initialising the set of candidates to just the upper bound $\theta_0$. Later on, in lines 14-16, new candidates are introduced using the candidate-suggesting function $q(\cdot)$ at a pace dictated by the growth function $g(t)$. Typically in aggregation schemes, each one of the base algorithms is run in isolation. However, there is nothing preventing us from making them share the data and as such, selecting a base algorithm in our case simply amounts to choosing the length scale

---

**Algorithm 1** Length scale Balancing GP-UCB (LB-GP-UCB)

---

**Require:** initial length scale value $\theta_0$; suspected regret bounds $R^\theta(\cdot)$;
    growth function $g(\cdot)$; confidence parameters $\{\xi_t\}_{t=1}^T$ and $\{\beta_t^\theta\}_{t=1}^T$
1: Set $\mathcal{D}_0 = \emptyset$, $\Theta_1 = \{\theta_0\}$, $S_0^\theta = \emptyset$ for all $\theta \in \Theta$, length scale counter $l = 1$
2: **for** $t = 1, \ldots, T$ **do**
3:     Select length scale $\theta_t = \arg\min_{\theta \in \Theta_t} R^\theta(|S_{t-1}^\theta| + 1)$
4:     Select point to query $\boldsymbol{x}_t = \arg\max_{\boldsymbol{x} \in \mathcal{X}} \mathrm{UCB}_{t-1}^{\theta_t}(\boldsymbol{x})$
5:     Query the black-box $y_t = f(\boldsymbol{x}_t) + \epsilon_t$
6:     Update data buffer $\mathcal{D}_t = \mathcal{D}_{t-1} \cup (x_t, y_t)$
7:     For each $\theta \in \Theta_t$, set $S_t^\theta = \{\tau = 1, \ldots, t : \theta_\tau = \theta\}$
8:     Initialise length scales set for new iteration $\Theta_{t+1} := \Theta_t$
9:     **if** $\forall_{\theta \in \Theta_t} |S_t^\theta| \neq 0$ **then**
10:        Define $L_t(\theta) = \left( \frac{1}{|S_t^\theta|} \sum_{\tau \in S_t^\theta} y_\tau - \sqrt{\frac{\xi_t}{|S_t^\theta|}} \right)$
11:        {# Eliminate underperforming length scale values}
12:        $\Theta_{t+1} = \left\{ \theta \in \Theta_t : L_t(\theta) + \frac{2}{|S_t^\theta|} \sum_{\tau \in S_t^\theta} \beta_\tau^\theta \sigma_{\tau-1}^\theta(\boldsymbol{x}_\tau) \geq \max_{\theta' \in \Theta_t} L_t(\theta') \right\}$
13:     **end if**
14:     **if** $q(l+1) \leq \frac{\theta_0}{g(t)}$ **then**
15:        $\Theta_{t+1} := \Theta_{t+1} \cup \{q(l+1)\}$ {# Add shorter length scales}
16:        $l := l+1$
17:     **end if**
18: **end for**

---

value we will use to fit the GP model at a given time step $t$, which is done in line 3. This choice is done by the regret-balancing rule $\theta_t = \arg\min_{\theta \in \Theta_t} R^\theta(|S_t^\theta| + 1)$, with $R^\theta(\cdot)$ defined as in Theorem 2.3 and $S_t^\theta$ being the set of iterations before $t$ at which length scale value $\theta$ was chosen. Note that we only need to know the scaling of the bound up to a constant. This rule implies that lower length scale values will be selected less frequently than higher values, as their regret bounds grow faster. After that, in line 4, the algorithm utilises the acquisition rule dictated by a model fitted with the selected length scale value to find the point to query next, $\boldsymbol{x}_t$. The idea is that, occasionally, $\theta_t$ will be set to one of the smaller values from $\Theta_t$ and, if that results in finding significantly better function values, then the rejection mechanism in lines 9-12 will remove longer length scales from the set of considered values, $\Theta_t$. Otherwise, we will keep all of the length scales and try again after some number of iterations. We now proceed to derive a regret bound for our developed algorithm.

## 4 Regret Bound and Proof Sketch

We now state the formal regret bound of the proposed algorithm, provide a brief sketch of the proof and discuss the result.

**Theorem 4.1.** *Let us use confidence parameters of* $\xi_t = 2\sigma_N^2 \log\left( d \ln(g(t)) \pi^2 t^2 \right) - \log 3\delta$ *and* $\beta_t^\theta = B(\theta, N^\star) + \sigma_N \sqrt{2(\gamma_{t-1}^\theta + 1 + \ln(2/\delta))}$, *then with probability at least* $1 - \delta$, *the cumulative regret* $R_T$ *of the Algorithm 1 admits the following bound:*

$$R_T = \mathcal{O}\left( (t_0 + \iota) B^\star + \left( R^{\theta^\star}(T) + \sqrt{T\xi_T} \right) \left( \left( \frac{\theta_0}{\theta^\star} \right)^d d \ln \frac{\theta_0}{\theta^\star} + \iota \right) \right),$$

*where* $t_0 = g^{-1}\left( e^{-1/d} \theta_0 / \theta^\star \right)$ *and* $\iota = d \ln g(T)$.

*Proof.* (sketch) We provide a sketch of the result here and defer the proof to Appendix D.

Let us denote by $t_0$ the iteration at which the first well-specified length scale ($\hat{\theta} \leq \theta^\star$) is added to the candidate set in line 15. This will happen at the first iteration after $g^{-1}(\frac{\theta_0}{\theta^\star})$, where the condition in line 14 will trigger. Given the ratios between consecutive candidates suggested by $q(\cdot)$, we get that

Table 1: Comparison of optimality for A-GP-UCB and LB-GP-UCB for fixed functions $g(\cdot)$ and $b(\cdot)$. $R^\star(T)$ refers to the scaling of the regret bound of an oracle optimiser, knowing the optimal hyperparameters. See Appendix H for more details.

| Algorithm | Optimality $R_T/R^\star(T)$ | |
| --- | --- | --- |
| | Unknown $\theta$ | Unknown $\theta$ and $B$ |
| A-GP-UCB [6] | $\mathcal{O}(g(T)^d)$ | $\mathcal{O}(b(T)g(T)^d)$ |
| LB-GP-UCB / LNB-GP-UCB (ours) | $\mathcal{O}(d\ln g(T))$ | $\mathcal{O}(d\ln g(T)\ln b(T))$ |

$t_0 = \lceil g^{-1}(\frac{\theta_0}{\theta^\star}e^{-1/d})\rceil$. On iterations up to $t_0$, we can potentially suffer the highest as possible, thus the cumulative regret can be bounded as:

$$R_T = \sum_{t=1,\ldots,t_0-1} r_t + \sum_{t=t_0,\ldots,T} r_t \leq 2B^\star t_0 + \tilde{R}_T,$$

where $\tilde{R}_T$ is the regret of the algorithm after $t_0$. Let us define by $\mathcal{T}$ the set of iterations, where we reject at least one length scale value in line 12. We thus have:

$$\tilde{R}_T = \sum_{t\in\mathcal{T}} r_t + \sum_{t\notin\mathcal{T}} r_t \leq 2B^\star|\mathcal{T}| + \sum_{t\notin\mathcal{T}} r_t \leq 2B^\star q^{-1}(\frac{\theta_0}{g(T)}) + \sum_{t\notin\mathcal{T}} r_t,$$

where the second inequality comes from the fact that we cannot reject more candidates than we have introduced in total. The remaining thing to do is to bound $\sum_{t\notin\mathcal{T}} r_t$. This expression is the cumulative regret of the iterations, where no candidates are rejected and where at least one of the well-specified candidates has been introduced. We can bound this term using a similar strategy as in [33]. First, we show that, with a probability of at least $1 - \delta$, the well-specified candidate introduced at $t_0$ will not be rejected. Second, since no other candidates are rejected at iterations $t \notin \mathcal{T}$, it means that the function values achieved at those iterations cannot be too different from the ones achieved when using $\hat{\theta}$. Using this fact, we arrive at a statement:

$$\sum_{t\notin\mathcal{T}} r_t \leq \left(R^{\hat{\theta}}(|S_t^{\hat{\theta}}|) + \sqrt{T\xi_T}\right)\left(\sum_{\theta\in\mathcal{M}_0}\sqrt{\frac{|S_t^\theta|}{|S_t^{\hat{\theta}}|}} + q^{-1}\left(\frac{\theta_0}{g(T)}\right)\right),$$

where $\mathcal{M}_0$ is the set of misspecified length scale values that were chosen at least once after $t_0$. The rest of the proof consists of bounding $\frac{|S_t^\theta|}{|S_t^{\hat{\theta}}|}$, which can be done due to the selection rule in line 3.

$\square$

**Optimality** In Appendix H, we show that for a fixed choice of growth function, we get $R_T/R^{\theta^\star}(T) = \mathcal{O}(d\ln g(T))$. This is an improvement compared to A-GP-UCB achieving $R_T/R^{\theta^\star}(T) = \mathcal{O}(g(T)^d)$. As such the bound of our algorithm is significantly closer to the optimal bound than the one of A-GP-UCB. The faster $g(\cdot)$ is increasing, the quicker we will be able to find the first well-specified candidate, which will decrease the term $t_0 B^\star$ in the bound. At the same time, it will increase all the terms depending on $g(T)$, but as our bound only scales with $d\log g(T)$, we are able to select much more aggressive growth functions than A-GP-UCB, whose bound scales with $g(T)^d$. In the Experiments section we compare the performance of LB-GP-UCB and A-GP-UCB using different growth functions $g(t)$ and show that the former algorithm is much more robust to the choice of $g(t)$.

**Extension to unknown** $N^\star$ As we discussed before, LB-GP-UCB requires us to know the initial RKHS norm $N^\star$. However, we can easily extend the algorithm to handle the case of unknown $N^\star$, which we do in Appendix F. In Algorithm 3 we present Length scale and Bound Balancing (LNB) — an algorithm, which in addition to having candidates for $\theta$ also maintains a number of candidates for $N^\star$. As such, it requires us to specify another growth function $b(t)$ for exploring new norm values as well as the initial RKHS norm $B_0$. We prove its cumulative regret bound in Theorem F.1. We can similarly derive the suboptimality gap for our algorithm in this case. We display it in Table 1 together with the gap of A-GP-UCB. We can see that in this setting, we also achieve an improvement.

# 5 Experiments

We now evaluate the performance of our algorithm on multiple synthetic and real-world functions. To run experiments we used the compute resources listed in Appendix I.1 and implemented based on the codebase of [2], which uses the BoTorch package [5, 34]. We open-source our code[2]. We used the UCB acquisition function and we compared different techniques for selecting the length scale value. For all experiments, we used isotropic $\nu$-Matérn kernel with $\nu = 2.5$. We standardise the observations before fitting the GP model and as such keep the kernel outputscale fixed to $1.0$. The first baseline we compare against is MLE, where the length scale value is optimised using a multi-start L-BFGS-B method [25] (the default BoTorch optimiser [5]) after each timestep by maximising the marginal likelihood for the data collected so far. The next baseline is MCMC, where we employ a fully Bayesian treatment of the unknown length scale value using the NUTS sampler [19], which we implemented using Pyro [7]. We use BoTorch's default hyperprior ($\theta \sim \Gamma(3, 6)$) and to select a new point we optimise the expected acquisition function under the posterior samples as described by [13]. We also compare against A-GP-UCB. To achieve a fair comparison, we used the same growth function $g(t) = \max\{t_0, \sqrt{t}\}$ for both LB-GP-UCB and A-GP-UCB across all experiments, where $t_0$ was selected so that at least 5 candidates are generated for $g(1)$. We study the impact of this choice in the ablation section. We used 10 initial points for each algorithm unless specified otherwise. To select upper bound $\theta_0$ for A-GP-UCB and LB-GP-UCB we fitted a length scale to initial data points with MLE (and we did not use MLE after that). We present the results in Figure 3 below. We show running times in Table 2 in Appendix I.2. We now describe each benchmark problem in detail.

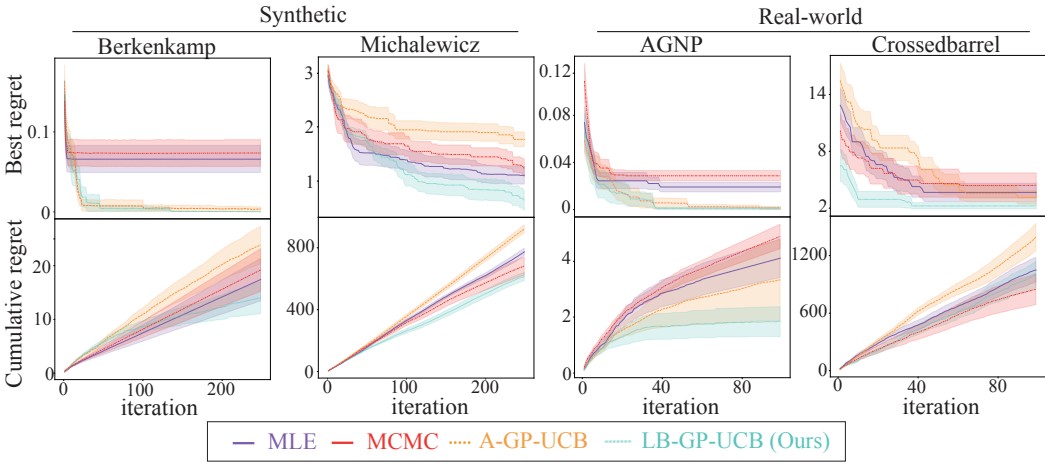

Figure 3: Regret results of the proposed algorithm and baselines on synthetic and real-world tasks. We ran 20 seeds on Berkenkamp and AGNP and 10 seeds on Michalewicz and Crossedbarrel problems. Shaded areas correspond to standard errors.

**Berkenkamp Toy Problem** We start with a one-dimensional toy problem proposed by the same paper that proposed the A-GP-UCB algorithm [6]. We showed a plot of this one-dimensional function in Figure 1. On the right side of the domain, the function appears to be smoother than on the left side. In this problem, we only use three initial points, to benchmark the ability of algorithms to escape from the local optimum on the right side of the domain. MLE and MCMC can be easily misled towards too-long length scale values, which causes them to get stuck in the local optimum. Both A-GP-UCB and LB-GP-UCB quickly find the optimal solution, however, due to over-exploration, the cumulative regret of A-GP-UCB grows faster than that of LB-GP-UCB.

**Michalewicz Synthetic Function** As a next benchmark, we evaluate our algorithm on the five-dimensional Michalewicz synthetic function, which has been designed to be challenging while using MLE for fitting hyperparameters, because it exhibits different degrees of smoothness throughout its domain. In the histogram in Figure 2, we compare the selected length scale value with an estimate of the optimal length scale $\hat{\theta}^*$. We produce this estimate by sampling ten thousand points uniformly through the domain and fitting a length scale value by maximum likelihood to the points with the

---

[2]https://github.com/JuliuszZiomek/LB-GP-UCB

top 1% of objective values. In this way, we are able to capture the length scale value that produces a good model for the Michalewicz function around the optimum, where it is least smooth. We can see LB-GP-UCB selects lengthscale value closer to $\hat{\theta}^*$ and as a result outperforms other baselines in terms of both cumulative and best regret metrics.

**Material Design Problems** We utilise material design tasks proposed by [17] and [30] - the 4-dimensional CrossedBarrel and 5-dimensional AGNP tasks. At each time step, the algorithm can choose which material configuration to try, and observe the objective value, which corresponds to a given material optimisation criterion. As material design problems are known to exhibit a needle-in-a-haystack behaviour [35], on both benchmarks MLE and MCMC get stuck at a suboptimal solution and their best regret does not fall beyond a certain value. A-GP-UCB is able to quickly find low-regret solutions on the AGNP benchmark, but struggles on the Crossedbarrel problem and underperforms in terms of cumulative regret. On the contrary, LB-GP-UCB performs well across both benchmark problems and across both regret metrics.

**Ablation on** $g(t)$ To test robustness of LB-GP-UCB, we evaluate it on Michalewicz function together with A-GP-UCB for different choices of $g(t)$. We try functions of form $g(t) = \max(t_0, t^a)$ for $a \in (0.25, 0.5, 0.75)$. In Figure 4 we plot the final performance of the algorithms after $N = 250$ steps as well as the distribution of selected length scale values. We see that A-GP-UCB is very sensitive to the selection of growth function $g(t)$, whereas our algorithm selects similar length scale values regardless of $g(t)$, which results in consistently good best regret results. We can also see that LB-GP-UCB typically selects values around $\hat{\theta}^*$ for different growth functions, whereas A-GP-UCB decreases its length scale beyond $\hat{\theta}^*$, resulting in slower convergence.

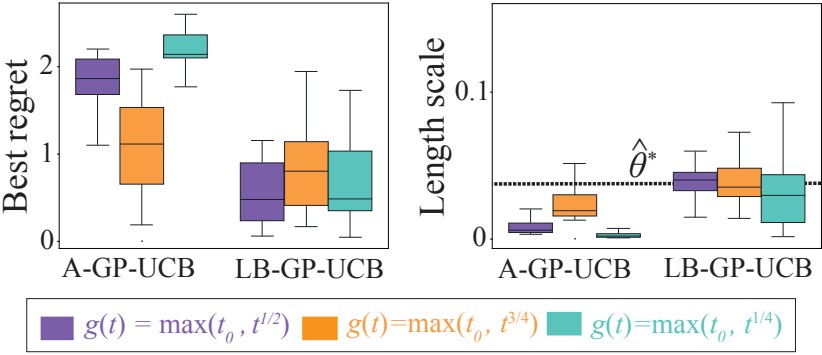

Figure 4: Ablation study of the choice of growth function $g(t)$. $t_0$ is chosen so that at least 5 candidates are generated at $g(1)$. See beginning of §5 for details.

# 6 Related Work

We already mentioned the work proposing A-GP-UCB [6], which so far has been the only work providing the guarantees on BO with unknown hyperparameters, where only an upper bound on the optimal length scale value is known. The work of [39] addresses the problem, where, in addition, a lower bound on the optimal length scale is known, however, the regret bound of the algorithm they propose scales with the $\gamma_T(k^{\theta_L})$ of the smallest possible length scale $\theta_L$, making it no better than a naïve algorithm always selecting $\theta_L$. [8] studied the problem of solving BO, when the kernel function is misspecified, however, provided no method for finding the well-specified kernel function. [26] proved a lower bound on the algorithm's regret in the case when the regularity of RKHS is unknown (which corresponds to an unknown $\nu$ hyperparameter in the case of Matérn kernel), compared to their work we focused on different unknown hyperparameters, such as the length scale. There have also been a number of works [13, 21, 27, 32] that tackled the problem of BO with unknown hyperparameters but did not provide a theoretical analysis of the used algorithm. Some of earlier works [15, 38] viewed BO with unknown hyperparameters as meta-learning or transfer learning problem, where a large dataset is available for pre-training. In our problem setting, we do not assume access to any such pre-training data. Within this work, we considered a frequentist problem setting, where the black-box function is arbitrarily selected from some RKHS. While there are no guarantees

for the consistency of MLE in such a setting, if we were to assume a Bayesian setting and put a GP prior on the black box, statistical literature derived asymptotic consistency results [4, 22, 24, 28] for MLE of kernel hyperparameters, including length scale. Under such a Bayesian setting, [41] studied the problem of BO with unknown prior, when we are given a finite number of candidate priors. [10] derived predictive guarantees for GP in a Bayesian setting with unknown hyperparameters, provided that hyperpriors on those hyperparameters are known. However, the authors do not provide any BO algorithm based on their results.

## 7    Conclusions

Within this work, we addressed the problem of BO with unknown hyperparameters. We proposed an algorithm with a cumulative regret bound logarithmically closer to optimal than the previous state of the art and showed that our algorithm can outperform existing baselines in practice. One limitation of our work is that we only showed how to handle the isotropic case, i.e. where the same length scale value is applied for every dimension. This limitation is because our algorithm requires the knowledge of the regret bounds of an optimiser utilising a given length scale value, which in turn requires the knowledge of the bounds on MIG $\gamma_T(k^\theta)$ for the used kernel. To the best of our knowledge, within the existing literature, no work has yet derived those bounds in non-isotropic cases. However, we believe that if such bounds were obtained, one could easily extend our algorithm to the non-isotropic case, in the same way as we extended our base algorithm to handle unknown norm and length scale simultaneously. This constitutes a promising direction of future work.

Another limitation of our work is the assumption of known noise magnitude $\sigma_N$. The problem of simultaneously not knowing kernel hyperparameters and noise magnitude is extremely challenging, as large variations in observed function values can be a result of either short lengthscale value or large noise magnitude. To the best of our knowledge, previous work did not tackle this setting and it remains an open problem.

Our algorithm relies on the standard GP model, which can result in poor scalability to large datasets and high-dimensional spaces. Extending our work to sparse GPs [29, 31] and kernels specifically designed for a high number of dimensions [14, 42] is another possible direction of future work.

## Acknowledgments and Disclosure of Funding

We would like to thank Toni Karvonen for pointing out important properties of the $\nu$-Matérn kernel's RKHS, which were crucial to take into account in the proposed problem setting. We would also like to thank Ondrej Bajgar and the anonymous reviewers for their helpful comments about improving the paper. Juliusz Ziomek was supported by the Oxford Ashton-Memorial Scholarship and EPSRC DTP grant EP/W524311/1. Masaki Adachi was supported by the Clarendon Fund, the Oxford Kobe Scholarship, the Watanabe Foundation, and Toyota Motor Corporation.

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

## A Proof of Proposition 2.2

**Proposition 2.2.** *We have that $\mathcal{I}_T(k^\theta) \leq \mathcal{O}\left(\gamma_T(k^\theta)\right)$:*

- *For an RBF kernel $\gamma_T(k^\theta) = \frac{1}{\theta^d} \log(T)^{d+1}$*

- *For $\nu$-Matérn kernel $\gamma_T(k^\theta) = \frac{1}{\theta^d} T^{\frac{d(d+1)}{2\nu+d(d+1)}} \log(T)^{\frac{2\nu}{2\nu+d}}$*

*Proof.* The RBF case follows directly from Proposition 2 in [6]. The same Proposition also provides a bound for the $\nu$-Matérn case, but more recent results allow us to derive a tighter bound. Section B.2. in [6] proves that the $\nu$-Matérn kernel has the $(C_p, \beta_P)$ polynomial eigendecay: $\beta_P = (2\nu + d)/d$ and $C_p = \left(\frac{1}{\theta}\right)^{2\nu+d}$, according to the definition of polynomial eigendecay as given by [37]. Substituting these values to Corollary 1 of [37], we get that in $\nu$-Matérn case

$$\mathcal{I}_T^\theta = \mathcal{O}\left(C_p^{1/\beta_P} T^{1/\beta_P} \log^{1-1/\beta_P}(T)^{1-1/\beta_P}\right) = \mathcal{O}\left(\left(\frac{1}{\theta}\right)^d T^{d/(2\nu+d)} \log^{2\nu/(d+2\nu)}(T)^{2\nu/(d+2\nu)}\right),$$

which finishes the proof. $\qquad\square$

## B General Hyperparameter case

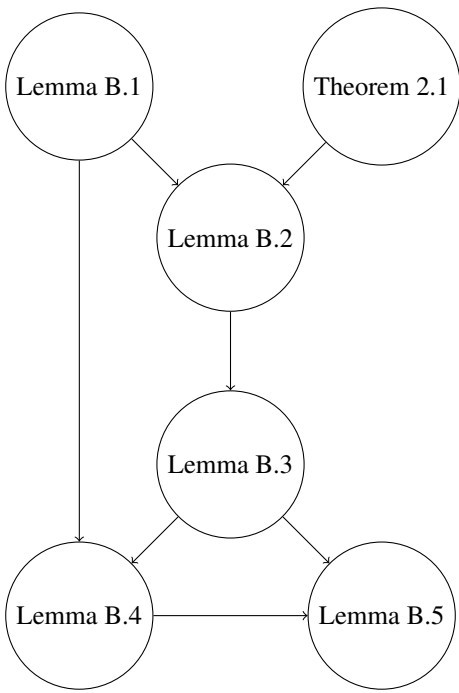

Figure 5: Diagram of the relationship between Lemmas in this Section. An incoming arrow means that the Lemma relies on the Lemma/ Theorem from which the arrow is outgoing. The final objective of this Section is to prove Lemma B.5.

We first derive a general result for a BO Algorithm utilising a regret-balancing scheme, under hyperparameters of any form. We present the pseudo-code of that procedure in Algorithm 2. We now proceed the prove the general regret bound for this algorithm. We would like to note that our proof closely follows the idea of [33]. We assume the unknown hyperparameter takes values in some set $\mathcal{U}$ and specifying this hyperparameter uniquely defines a kernel function $k^u(\cdot, \cdot)$ and the RKHS norm $B^u$. We say a hyperparameter value is well-specified if $|f|_{k^u} \leq B^u$. Algorithm 2 requires hyperparameter-proposing function $a(\cdot)$ as one of the inputs, which optionally expands the set of considered hyperparameters. We will denote by $A = \bigcup_{t=0,\ldots,T} a(t)$ the set of all hyperparameters

introduced by $a(\cdot)$ up to step $T$. We will also denote $\mathcal{W}$ to mean the set of all well-specified hyperparameters in $A$ and $\mathcal{M} = A \setminus \mathcal{W}$ to denote the set of misspecified hyperparameters. Let $u^\star$ be the first well-specified hyperparameter introduced. We will also write $\mathcal{T}$ to mean the set of all iterations after (and including) $t_0$, where at least one hyperparameter value was rejected, that is:

$$\mathcal{T} = \left\{ t = t_0, \ldots, T \middle| \exists_{u \in U_t} L_t(u) + \frac{2}{|S_t^u|} \sum_{\tau \in S_t^u} \beta_\tau^u \sigma_{\tau-1}^u(\boldsymbol{x}_\tau) < \max_{u' \in U_t} L_t(u') \right\}.$$

We also define $\mathcal{M}_0 = \{u \in \mathcal{M} | \exists_{t \geq t_0} u_t = u\}$ to be the set of all misspecified hyperparameters that were selected at least once after $t_0$.

---

**Algorithm 2** Hyperparameter Balancing GP-UCB

---

**Require:** suspected regret bounds $R^u(\cdot)$;
hyperparameter-proposing function $a(\cdot)$; confidence parameters $\{\xi_t\}_{t=1}^T$ and $\{\beta_t^u\}_{t=1}^T$
1: Set $\mathcal{D}_0 = \emptyset$, $U_1 = a(0)$, $S_0^u = \emptyset$ for all $u \in U_1$,
2: **for** $t = 1, \ldots, T$ **do**
3:     Select hyperparameter $u_t = \arg\min_{u \in U_t} R^u(|S_{t-1}^u| + 1)$
4:     Select point to query $\boldsymbol{x}_t = \arg\max_{\boldsymbol{x} \in \mathcal{X}} \text{UCB}_{t-1}^{u_t}(\boldsymbol{x})$
5:     Query the black-box $y_t = f(\boldsymbol{x}_t)$
6:     Update data buffer $\mathcal{D}_t = \mathcal{D}_{t-1} \cup (x_t, y_t)$
7:     For each $u \in U_t$, set $S_t^u = \{\tau = 1, \ldots, t : u_\tau = u\}$
8:     Initialise hyperparameter set for new iteration $U_{t+1} := U_t$
9:     **if** $\forall_{u \in U_t} |S_t^u| \neq 0$ **then**
10:         Define $L_t(u) = \left( \frac{1}{|S_t^u|} \sum_{\tau \in S_t^u} y_\tau - \sqrt{\frac{\xi_t}{|S_t^u|}} \right)$
11:         $U_{t+1} = \left\{ u \in U_t : L_t(u) + \frac{2}{|S_t^u|} \sum_{\tau \in S_t^u} \beta_\tau^u \sigma_{\tau-1}^u(\boldsymbol{x}_\tau) \geq \max_{u' \in U_t} L_t(u') \right\}$
12:     **end if**
13:     Optionally expand hyperparameter set $U_{t+1} := U_{t+1} \cup \{a(t)\}$
14: **end for**

---

We will first provide some auxiliary Lemmas that we will be required to prove the general result for Algorithm 2. We first recall a result from existing literature on the concentration of noise.

**Lemma B.1** (Lemma 5.1 in [41]). *For each $u \in A$ and $t = 1, \ldots, T$ we have:*

$$P \left( \forall_{t=1,\ldots,T} \ \forall_{u \in A} \left| \sum_{i \in S_t^u} \epsilon_i \right| \leq \sqrt{\xi_t |S_t^u|} \right) \geq 1 - \delta_B,$$

*where $\xi_t = 2\sigma_N^2 \log \frac{|A|\pi^2 t^2}{6\delta_B}$.*

The next result we will need is the high-probability guarantee that the optimal hyperparameter $u^\star$ will not be removed from the set of considered hyperparameters after it is introduced.

**Lemma B.2.** *If the events of Theorem 2.1 and Lemma B.1 hold, then the optimal hyperparameter $u^*$ is never removed after it is introduced, i.e. $u^* \in U_t$ for all $t = t_0, \ldots, T$, where $t_0$ is the first iteration, where the optimal hyperparameter $u^\star$ is introduced.*

*Proof.* First, observe that under these events, the optimal hyperparameter $u^*$ will never be removed. To see this, observe that at any time set $t$ such that $u_t = u^\star$ the following holds:

$$f(\boldsymbol{x}^*) - f(\boldsymbol{x}_t) \leq \mu_t^{u^\star}(\boldsymbol{x}^\star) + \beta_t^{u^\star} \sigma_t^{u^\star}(\boldsymbol{x}^\star) - \mu_t^{u^\star}(\boldsymbol{x}_t) + \beta_t^{u^\star} \sigma_t^{u^\star}(\boldsymbol{x}_t)$$

$$\leq \mu_t^{u^\star}(\boldsymbol{x}_t) + \beta_t^{u^\star} \sigma_t^{u^\star}(\boldsymbol{x}_t) - \mu_t^{u^\star}(\boldsymbol{x}_t) + \beta_t^{u^\star} \sigma_t^{u^\star}(\boldsymbol{x}_t)$$

$$= 2\beta_t^{u^\star} \sigma_t^{u^\star}(\boldsymbol{x}_t),$$

where the first inequality follows from the fact that the event of Theorem 2.1 holds and $u^\star$ is well-specified, and the second inequality is because we chose $\boldsymbol{x}_t$ so as to maximise the UCB. Using this

fact, we get that:

$$\frac{1}{|S_t^{u^*}|} \sum_{t \in S_t^{u^*}} (f(\boldsymbol{x}^*) - (y_t - \epsilon_t)) = \frac{1}{|S_t^{u^*}|} \sum_{t \in S_t^{u^*}} (f(\boldsymbol{x}^*) - f(\boldsymbol{x}_t)) \leq \frac{2}{|S_t^{u^*}|} \sum_{t \in S_t^{u^*}} \beta_t^{u^*} \sigma_t^{u^*}(\boldsymbol{x}_t)$$

Rearranging, we get:

$$f(\boldsymbol{x}^*) \leq \frac{2}{|S_t^{u^*}|} \sum_{t \in S_t^{u^*}} \beta_t^{u^*} \sigma_t^{u^*}(\boldsymbol{x}_t) - \frac{1}{|S_t^{u^*}|} \sum_{t \in S_t^{u^*}} \epsilon_t + \sum_{t \in S_t^{u^*}} \frac{y_t}{|S_t^{u^*}|}$$

$$\leq \frac{2}{|S_t^{u^*}|} \sum_{t \in S_t^{u^*}} \beta_t^{u^*} \sigma_t^{u^*}(\boldsymbol{x}_t) + \sqrt{\frac{\xi_t}{|S_t^{u^*}|}} + \sum_{t \in S_t^{u^*}} \frac{y_t}{|S_t^{u^*}|}, \tag{1}$$

where the second inequality is a result of Lemma B.1. Now, for any $u \in A$ and any $t \in S_T^u$, we have the following:

$$f(\boldsymbol{x}^*) = \frac{1}{|S_t^u|} \sum_{t \in S_t^u} f(\boldsymbol{x}^*) \geq \frac{1}{|S_t^u|} \sum_{t \in S_t^u} f(\boldsymbol{x}_t)$$

$$= \sum_{t \in S_t^u} \frac{y_t}{|S_t^u|} - \sum_{t \in S_t^u} \frac{\epsilon_t}{|S_t^u|} \geq \sum_{t \in S_t^u} \frac{y_t}{|S_t^u|} - \sqrt{\frac{\xi_t}{|S_t^u|}}, \tag{2}$$

where again second inequality comes from Lemma B.1. Thus combining Inequalities 1 and 2, we get that for any $u \in A$:

$$\frac{2}{|S_t^{u^*}|} \sum_{t \in S_t^{u^*}} \beta_t^{u^*} \sigma_t^{u^*}(\boldsymbol{x}_t) + \sqrt{\frac{\xi_t}{|S_t^{u^*}|}} + \sum_{t \in S_t^{u^*}} \frac{y_t}{|S_t^{u^*}|} \geq \sum_{t \in S_t^u} \frac{y_t}{|S_t^u|} - \sqrt{\frac{\xi_t}{|S_t^u|}},$$

which means the `if` statement in line 11 will always evaluate to `true` for $u^*$ and thus $u^* \in U_t$ for all $t = 1, \ldots, T$. □

We now recall the balancing condition, which is satisfied by regret balancing algorithms as proven in Lemma 5.2 of [33]. This condition basically says that due to the selection rule that chooses an algorithm with current lower suspected regret, the regret of any two algorithms (that have not been rejected) must be "close" to each other. We prove a slightly different statement than that in [33], as in our case the set of base algorithms can be dynamically expanded.

**Lemma B.3.** *Assume the event of Lemma B.2 holds. For any $u \in A$ that was selected in line 3 at least once after $t_0$ and any time step $t = t_0, \ldots, T$ we must have that $R^u(|S_t^u|) \leq R^{u^*}(|S_t^{u^*}|) + 2B^{u^*}$.*

*Proof.* We will prove the statement by contradiction. If the statement of the Lemma was not true, we would have:

$$R^u(|S_t^u|) > R^{u^*}(|S_t^{u^*}|) + 2B^{u^*} \geq R^{u^*}(|S_t^{u^*}| + 1),$$

where the second inequality comes from the assumption that the bounds are non-trivial. If $u$ has been selected at least once after $u^*$ was introduced (and by Lemma B.2 $u^*$ could not have been excluded afterwards), then the last time $u$ was selected, $u^*$ must have been present in $U_t$ and as such selection of $u$ such that $R^u(|S_t^u|) > R^{u^*}(|S_t^{u^*}| + 1)$ would violate the selection rule in line 3. □

Before we can prove the final result of this Section, we need one more auxiliary Lemma. This Lemma bounds the regret on all the iterations, where none of the hyperparameters was rejected (i.e. $t \notin \mathcal{T}$), by using the rejection rule of line 11.

**Lemma B.4.** *Assume the event of Lemma B.3 holds. Let us call by $\mathcal{T}$ the set of all iterations after $t_0$, where we do not reject any of the candidates. We must have that for any $u \in A$:*

$$\sum_{\substack{t \notin \mathcal{T} \\ t \in S_T^u}} r_t \leq \left(\frac{|S_T^u|}{|S_T^{u^*}|} + 1\right) CR^{u^*}(|S_T^{u^*}|) + 2CB^{u^*} + 2\left(\sqrt{\frac{|S_T^u|}{|S_T^{u^*}|}} + 1\right) \sqrt{|S_T^u| \xi_t}.$$

*Proof.* Let $t'$ be the smallest iteration after $t_0$, such that $S_{t'}^u = S_T^u$ and $t' \notin \mathcal{T}$, i.e. $t'$ is the last iteration not in $\mathcal{T}$ when $u$ was played. Notice that since $t' \notin \mathcal{T}$, all hyperparameter values in $U_{t'}$ must satisfy the `if` statement in line 11 when compared with any other hyperparameter value in $U_{t'}$. Let us choose any hyperparameter value in $U_{t'}$ and compare it with $u^\star$ (which must be in $U_{t'}$ due to the event of Lemma B.3 that guarantees preservation of $u^\star$.). For the `if` statement to evaluate to `true`, we must have:

$$\frac{1}{|S_{t'}^u|} \sum_{\substack{t \notin \mathcal{T} \\ t \in S_{t'}^u}} y_t + \frac{2}{|S_{t'}^u|} \sum_{t \in S_{t'}^u} \beta_t^u \sigma_t^u(\boldsymbol{x}_t) + \sqrt{\frac{\xi_t}{|S_{t'}^u|}} \geq \frac{1}{|S_t^{u^*}|} \sum_{t \in S_t^{u^*}} y_t - \sqrt{\frac{\xi_t}{|S_t^{u^*}|}}.$$

Multiplying both sides by $-1$ and adding $f(\boldsymbol{x}^*)$, we get:

$$\frac{1}{|S_{t'}^u|} \sum_{\substack{t \notin \mathcal{T} \\ t \in S_{t'}^u}} (f(\boldsymbol{x}^*) - y_t) - \frac{2 \sum_{t \in S_{t'}^u} \beta_t^u \sigma_t^u(\boldsymbol{x}_t)}{|S_{t'}^u|} - \sqrt{\frac{\xi_t}{|S_{t'}^u|}} \leq \frac{1}{|S_{t'}^{u^*}|} \sum_{t \in S_{t'}^{u^*}} (f(\boldsymbol{x}^*) - y_t) + \sqrt{\frac{\xi_t}{|S_{t'}^{u^*}|}}.$$

Using the fact that $y_t = f(\boldsymbol{x}_t) + \epsilon_t$ and Lemma B.1 we get that:

$$\frac{1}{|S_{t'}^u|} \sum_{\substack{t \notin \mathcal{T} \\ t \in S_{t'}^u}} (f(\boldsymbol{x}^*) - f(\boldsymbol{x}_t)) - \frac{2 \sum_{t \in S_{t'}^u} \beta_t^u \sigma_t^u(\boldsymbol{x}_t)}{|S_{t'}^u|} - 2\sqrt{\frac{\xi_t}{|S_{t'}^u|}}$$

$$\leq \frac{1}{|S_{t'}^{u^\star}|} \sum_{t \in S_{t'}^{u^\star}} (f(\boldsymbol{x}^*) - f(\boldsymbol{x}_t)) + 2\sqrt{\frac{\xi_t}{|S_{t'}^{u^\star}|}}.$$

Rearranging and observing that $r_t = f(\boldsymbol{x}^\star) - f(\boldsymbol{x}_t)$, we obtain:

$$\sum_{\substack{t \notin \mathcal{T} \\ t \in S_{t'}^u}} r_t \leq 2 \sum_{t \in S_{t'}^u} \beta_t^u \sigma_t^u(\boldsymbol{x}_t) + 2\sqrt{|S_{t'}^u| \xi_t} + \frac{|S_{t'}^u|}{|S_{t'}^{u^\star}|} \sum_{t \in S_{t'}^{u^\star}} r_t + 2|S_{t'}^u| \sqrt{\frac{\xi_t}{|S_{t'}^{u^\star}|}}$$

$$\leq CR^u(|S_{t'}^u|) + 2\sqrt{|S_{t'}^u| \xi_t} + \frac{|S_{t'}^u|}{|S_{t'}^{u^\star}|} CR^{u^\star}(|S_{t'}^{u^\star}|) + 2|S_{t'}^u| \sqrt{\frac{\xi_t}{|S_{t'}^{u^\star}|}},$$

where we use the fact that $2 \sum_{t \in S_{t'}^u} \beta_t^u \sigma_t^u(\boldsymbol{x}_t) \leq CR^u(S_{t'}^u)$ for some constant $C > 0$, which follows from the proof of Theorem 2 of [11] on which we rely to obtain the suspected regret bounds. We also used the fact that due to $u^\star$ being well-specified we have that $\sum_{t \in S_T^{u^\star}} r_t \leq CR^{u^\star}(|S_T^{u^\star}|)$. We now apply Lemma B.3 to get that $R^u(|S_t^u|) \leq R^{u^\star}(|S_t^{u^\star}|) + 2B^{u^\star}$. Substituting that to the bound developed above and noting that by definition $S_{t'}^u = S_T^u$, we get:

$$\sum_{\substack{t \notin \mathcal{T} \\ t \in S_T^u}} r_t = \sum_{\substack{t \notin \mathcal{T} \\ t \in S_{t'}^u}} r_t \leq \left( \frac{|S_{t'}^u|}{|S_{t'}^{u^*}|} + 1 \right) CR^{u^\star}(|S_{t'}^{u^*}|) + 2CB^{u^\star} + 2 \left( \sqrt{\frac{|S_{t'}^u|}{|S_{t'}^{u^*}|}} + 1 \right) \sqrt{|S_{t'}^u| \xi_t}$$

$$= \left( \frac{|S_T^u|}{|S_T^{u^*}|} + 1 \right) CR^{u^\star}(|S_T^{u^*}|) + 2CB^{u^\star} + 2 \left( \sqrt{\frac{|S_T^u|}{|S_T^{u^*}|}} + 1 \right) \sqrt{|S_T^u| \xi_t},$$

which finishes the proof. $\qquad \square$

We are now ready to prove the final result of this Section.

**Lemma B.5.** *Let us run Algorithm 2 for $T$ iterations with a given choice of the hyperparameter-proposing function $a : \mathbb{N} \to \mathcal{U}$. Let $\mathcal{T}$ be the set of all iterations after $t_0$, where at least one hyperparameter is rejected by operation in line 11. If we set $\xi_t = 2\sigma_N^2 \log \frac{|A| \pi^2 t^2}{6 \delta_B}$ and $\beta_t^u = B^u + \sigma_N \sqrt{2(\gamma_{t-1}^u + 1 + \ln(1/\delta_A))}$, we then have that with probability as least $1 - \delta_A - \delta_B$:*

$$\sum_{t \notin \mathcal{T}} r_t = \mathcal{O} \left( |A| B^{u^*} + \left( R^{u^*}(T) + \sqrt{T \xi_T} \right) \left( \sum_{u \in \mathcal{M}_0} \sqrt{\frac{|S_t^u|}{|S_{t'}^{u^*}|}} + |A| \right) \right).$$

*Proof.* We will prove the bound assuming the events of Theorem 2.1 and Lemma B.1 hold. As this happens with probability at least $1 - \delta_A - \delta_B$, the bound also holds with the same probability. We have that:

$$
\begin{aligned}
\sum_{t \notin \mathcal{T}} r_t &\leq \sum_{u \in \mathcal{W}} \sum_{\substack{t \notin \mathcal{T} \\ t \in S_T^u}} r_t + \sum_{u \in \mathcal{M}_0} \sum_{\substack{t \notin \mathcal{T} \\ t \in S_T^u}} r_t \\
&\leq \sum_{u \in \mathcal{W}} CR^u(|S_T^u|) + \sum_{u \in \mathcal{M}_0} \sum_{\substack{t \notin \mathcal{T} \\ t \in S_T^u}} r_t \\
&\leq \sum_{u \in \mathcal{W}} \left( CR^{u^*}(|S_T^u|) + 2CB^{u^*} \right) + \sum_{u \in \mathcal{M}_0} \sum_{\substack{t \notin \mathcal{T} \\ t \in S_T^u}} r_t \\
&\leq |\mathcal{W}| \left( CR^{u^*}(T) + 2CB^{u^*} \right) + \sum_{u \in \mathcal{M}_0} \sum_{\substack{t \notin \mathcal{T} \\ t \in S_T^u}} r_t, \quad (3)
\end{aligned}
$$

where the first transition is due to all hyperparameters in $\mathcal{W}$ being well-specified and the second transition is due to Lemma B.3. We now tackle the second term:

$$
\begin{aligned}
\sum_{u \in \mathcal{M}_0} \sum_{\substack{t \notin \mathcal{T} \\ t \in S_T^u}} r_t &\leq \sum_{u \in \mathcal{M}_0} \left( \left( \frac{|S_T^u|}{|S_T^{u^*}|} + 1 \right) CR^{u^*}(|S_T^{u^*}|) + 2CB^{u^*} + 2 \left( \sqrt{\frac{|S_T^u|}{|S_T^{u^*}|}} + 1 \right) \sqrt{|S_T^u| \xi_t} \right) \\
&\leq \left( \sum_{u \in \mathcal{M}_0} \frac{|S_T^u|}{|S_T^{u^*}|} + |\mathcal{M}_0| \right) CR^{u^*}(|S_T^{u^*}|) + 2C|\mathcal{M}_0| B^{u^*} + 2 \left( \sum_{u \in \mathcal{M}_0} \sqrt{\frac{|S_T^u|}{|S_T^{u^*}|}} + |\mathcal{M}_0| \right) \sqrt{T \xi_t} \\
&= \mathcal{O} \left( \left( \sum_{u \in \mathcal{M}_0} \frac{|S_T^u|}{|S_T^{u^*}|} + |\mathcal{M}_0| \right) R^{u^*}(|S_T^{u^*}|) + |\mathcal{M}_0| B^{u^*} + \left( \sum_{u \in \mathcal{M}_0} \sqrt{\frac{|S_T^u|}{|S_T^{u^*}|}} + |\mathcal{M}_0| \right) \sqrt{T \xi_t} \right),
\end{aligned}
$$

where we used Lemma B.4, Cauchy-Schwarz inequality and the fact that $|S_T^u| \leq T$ for all $u \in A$. Observe, that suspected regret bounds in BO will be of form $R^u(T) = \sqrt{T \beta_T^u \gamma_T^u} (\sqrt{\gamma_T^u} + \sqrt{B^u})$. Substituting this fact, we get:

$$
\begin{aligned}
R^{u^*}(|S_T^{u^*}|) \sum_{u \in \mathcal{M}_0} \frac{|S_T^u|}{|S_T^{u^*}|} &\leq \mathcal{O} \left( \sum_{u \in \mathcal{M}_0} \frac{|S_T^u|}{|S_T^{u^*}|} \sqrt{|S_T^{u^*}| \beta_T^{u^*} \gamma_T^u} \left( \sqrt{\gamma_T^{u^*}} + \sqrt{B^{u^*}} \right) \right) \\
&= \mathcal{O} \left( \sum_{u \in \mathcal{M}_0} \sqrt{\frac{|S_T^u|}{|S_T^{u^*}|}} \sqrt{|S_T^u| \beta_T^{u^*} \gamma_T^u} \left( \sqrt{\gamma_T^{u^*}} + \sqrt{B^{u^*}} \right) \right) \\
&= \mathcal{O} \left( \sum_{u \in \mathcal{M}_0} \sqrt{\frac{|S_T^u|}{|S_T^{u^*}|}} R^{u^*}(|S_T^u|) \right) \\
&\leq \mathcal{O} \left( \sum_{u \in \mathcal{M}_0} \sqrt{\frac{|S_T^u|}{|S_T^{u^*}|}} R^{u^*}(T) \right).
\end{aligned}
$$

We thus get :

$$
\sum_{u \in \mathcal{M}_0} \sum_{\substack{t \notin \mathcal{T} \\ t \in S_T^u}} r_t = \mathcal{O} \left( |\mathcal{M}_0| B^{u^*} + \left( R^{u^*}(T) + \sqrt{T \xi_T} \right) \left( \sum_{u \in \mathcal{M}_0} \sqrt{\frac{|S_T^u|}{|S_T^{u^*}|}} + |\mathcal{M}_0| \right) \right).
$$

Substituting this back into Equation 3 yields the following bound:

$$
\sum_{t \notin \mathcal{T}} r_t \leq \mathcal{O} \left( |A| B^{u^*} + \left( R^{u^*}(T) + \sqrt{T \xi_T} \right) \left( \sum_{u \in \mathcal{M}_0} \sqrt{\frac{|S_T^u|}{|S_T^{u^*}|}} + |A| \right) \right)
$$

$\square$

## C  Proof of Lemma 3.1

**Lemma 3.1.** *In the case of both RBF and $\nu$-Matérn kernel, we have that:*

$$\frac{R^{\hat{\theta}}(T)}{R^{\theta^\star}(T)} = \mathcal{O}\left(1\right).$$

*Proof.*

$$\frac{R^{\hat{\theta}}(T)}{R^{\theta^\star}(T)} = \frac{\sqrt{T\gamma_T^{\hat{\theta}}}\left(\sqrt{\gamma_T^{\hat{\theta}}} + B(\hat{\theta}, N^\star)\right)}{\sqrt{T\gamma_T^{\theta^\star}}\left(\sqrt{\gamma_T^{\theta^\star}} + B(\theta^\star, N^\star)\right)} = 2\left(\frac{\theta^\star}{\hat{\theta}}\right)^d \leq 2\left(\frac{q(i^\star)}{q(i^\star+1)}\right)^d = 2\left(e^{-1/d}\right)^d = \mathcal{O}(1),$$

where $i^\star = \max\{i \in \mathbb{N} \mid q(i) \geq \theta^\star\}$ and as such we have $q(i^\star + 1) = \hat{\theta} \leq \theta^\star \leq q(i^\star)$. $\qquad\square$

## D  Proof of Theorem 4.1

**Theorem 4.1.** *Let us use confidence parameters of $\xi_t = 2\sigma_N^2 \log\left(d\ln(g(t))\pi^2 t^2\right) - \log 3\delta$ and $\beta_t^\theta = B(\theta, N^\star) + \sigma_N\sqrt{2(\gamma_{t-1}^\theta + 1 + \ln(2/\delta))}$, then with probability at least $1 - \delta$, the cumulative regret $R_T$ of the Algorithm 1 admits the following bound:*

$$R_T = \mathcal{O}\left((t_0 + \iota)B^\star + \left(R^{\theta^\star}(T) + \sqrt{T\xi_T}\right)\left(\left(\frac{\theta_0}{\theta^\star}\right)^d d\ln\frac{\theta_0}{\theta^\star} + \iota\right)\right),$$

*where $t_0 = g^{-1}\left(e^{-1/d}\theta_0/\theta^\star\right)$ and $\iota = d\ln g(T)$.*

*Proof.* We start with a similar regret decomposition as in the proof of Theorem 1 in [6]. Let $t_0$ be the first iteration, where a length scale value smaller or equal to $\theta^\star$ enters the hyperparameter set $\Theta_{t_0}$. We will refer to that value as $\hat{\theta}$. Before this happens, all hyperparameters in the set are misspecified and as such we cannot guarantee anything about the regret of those iterations. As such, we bound their regret by $2B^\star$, which is the highest possible regret one can suffer at one iteration. We thus get:

$$R_T = \sum_{t=1,\ldots,t_0} r_t + \sum_{t=t_0+1,\ldots,T} r_t \leq t_0 2B^\star + \tilde{R}_T \tag{4}$$

We note that due to how we add new length scales, we have that $t_0 \leq g^{-1}(\frac{\theta_0}{\theta^\star e^{1/d}})$ and we defined $\tilde{R}_T$ be the cumulative regret of all iterations after $t_0$. Let us define the set $\mathcal{T}$ to be the set of all iterations, where at least one hyperparameter was eliminated. We thus get the following regret bound:

$$\tilde{R}_T = \sum_{t\in\mathcal{T}} r_t + \sum_{t\notin\mathcal{T}} r_t \leq 2|\mathcal{T}|B^\star + \sum_{t\notin\mathcal{T}} r_t \leq 2q^{-1}\left(\frac{\theta_0}{g(T)}\right)B^\star + \sum_{t\notin\mathcal{T}} r_t,$$

where the last inequality comes from the fact that we cannot reject more hyperparameters than we have considered in total. We now rely on Lemma B.5, which provides a bound on $\sum_{t\notin\mathcal{T}} r_t$ with probability at least $1 - \delta_A - \delta_B$ and we set $\delta_A = \delta_B = \delta/2$. In the notation of the Lemma, we can write $A = \bigcup_{t=1,\ldots,T}\Theta_t$ to mean the set of all length scale values introduced over the course of the algorithm running and by $\mathcal{M}_0$ we mean all length scales longer than $\hat{\theta}$ that were selected at least once after $t_0$. We observe that $|A| \leq q^{-1}\left(\frac{\theta_0}{g(T)}\right)$ and our optimal base learner is $u^\star = \hat{\theta}$. This gives us:

$$\sum_{t\notin\mathcal{T}} r_t = \mathcal{O}\left(q^{-1}\left(\frac{\theta_0}{g(T)}\right)B^\star + \left(R^{\hat{\theta}}(T) + \sqrt{T\xi_T}\right)\left(\sum_{u\in\mathcal{M}_0}\sqrt{\frac{|S_T^u|}{|S_T^{u^*}|}} + q^{-1}\left(\frac{\theta_0}{g(T)}\right)\right)\right)$$

To finish the proof we rely on the following Lemma, which we prove in Appendix E.

**Lemma D.1.** *If the event of Lemma B.5 holds, then for any $\theta \in \mathcal{M}_0$ and $t \geq t_0$ we have that* $\sqrt{\frac{|S_t^\theta|}{|S_t^{\theta^\star}|}} \leq \left(\frac{\theta_0}{\theta^\star}\right)^d.$

We thus get the following final bound:

$$\sum_{t \notin \mathcal{T}} r_t = \mathcal{O}\left( q^{-1}\left(\frac{\theta_0}{g(T)}\right) B^\star + \left(R^{\hat{\theta}}(T) + \sqrt{T\xi_T}\right)\left(|\mathcal{M}_0|\left(\frac{\theta_0}{\theta^\star}\right)^d + q^{-1}\left(\frac{\theta_0}{g(T)}\right)\right)\right).$$

We now observe that $|\mathcal{M}_0| = \mathcal{O}\left(q^{-1}(\theta^\star)\right) = \mathcal{O}\left(d \ln \frac{\theta_0}{\theta^\star}\right)$. We substitute the bound above to Equation 4, together with bound on $|\mathcal{M}_0|$ to obtain:

$$R_T \leq \mathcal{O}\left(\left(g^{-1}\left(\frac{\theta_0}{\theta^\star e^{1/d}}\right) + \iota\right) B^\star + \left(R^{\hat{\theta}}(T) + \sqrt{T\xi_T}\right)\left(\left(\frac{\theta_0}{\theta^\star}\right)^d d \ln \theta^\star + \iota\right)\right),$$

where $\iota = d \ln g(T)$. By Lemma 3.1 we know we can just replace $R^{\hat{\theta}}(T)$ with $R^{\theta^\star}(T)$ in the bound above, which finishes the proof.

$\square$

# E   Proof of Lemma D.1

**Lemma D.1.** *If the event of Lemma B.5 holds, then for any $\theta \in \mathcal{M}_0$ and $t \geq t_0$ we have that* $\sqrt{\frac{|S_t^\theta|}{|S_t^{\theta^\star}|}} \leq \left(\frac{\theta_0}{\theta^\star}\right)^d.$

*Proof.* If $|S_t^{\theta^\star}| \geq |S_t^\theta|$, the bound holds trivially. Thus we will assume $|S_t^{\theta^\star}| < |S_t^\theta|$. The suspected regret bounds are of the form:

$$R^\theta(T) = \sqrt{T\gamma_T^\theta}\left(\sqrt{\gamma_T^\theta} + B(\theta, N^\star)\right).$$

If the event on Lemma B.5 holds that means the event of Lemma B.3 holds as well. Due to the regret balancing condition from Lemma B.3 and the non-triviality of bounds, we have:

$$R^\theta(|S_t^\theta|) \leq R^{\theta^\star}(|S_t^{\theta^\star}|) + 2B^{u^\star} \leq 2R^{\theta^\star}(|S_t^{\theta^\star}|)$$

$$\sqrt{\frac{|S_t^\theta|}{|S_t^{\theta^\star}|}} \leq 2 \frac{\sqrt{\gamma_{|S_t^{\theta^\star}|}^\theta}\left(\sqrt{\gamma_{|S_t^{\theta^\star}|}^\theta} + B(\theta^\star, N^\star)\right)}{\sqrt{\gamma_{|S_t^\theta|}^\theta}\left(\sqrt{\gamma_{|S_t^\theta|}^\theta} + B(\theta, N^\star)\right)}.$$

To finish the proof we consider the following two cases.

**Case 1:** Consider the case when $\sqrt{\gamma_{|S_t^{\theta^\star}|}^\theta} \geq B(\theta^\star, N^\star)$. We then have:

$$\sqrt{\frac{|S_t^\theta|}{|S_t^{\theta^\star}|}} \leq 2 \frac{\gamma_{|S_t^{\theta^\star}|}^\theta}{\gamma_{|S_t^\theta|}^\theta} \leq 2\left(\frac{\theta}{\theta^\star}\right)^d \frac{\gamma_{|S_t^\theta|}^\theta}{\gamma_{|S_t^\theta|}^\theta} = 2\left(\frac{\theta}{\theta^\star}\right)^d \leq 2\left(\frac{\theta_0}{\theta^\star}\right)^d.$$

**Case 2** Consider the case when $\sqrt{\gamma_{|S_t^{\theta^\star}|}^\theta} < B(\theta^\star, N^\star)$. We then have:

$$\sqrt{\frac{|S_t^\theta|}{|S_t^{\theta^\star}|}} \leq 2 \frac{\sqrt{\gamma_{|S_t^{\theta^\star}|}^\theta} B(\theta^\star, N^\star)}{\sqrt{\gamma_{|S_t^\theta|}^\theta} B(\theta, N^\star)} \leq 2\left(\frac{\theta}{\theta^\star}\right)^{d/2} \sqrt{\frac{\gamma_{|S_t^\theta|}^\theta}{\gamma_{|S_t^\theta|}^\theta}} \left(\frac{\theta}{\theta^\star}\right)^{d/2} \frac{B(\theta, N^\star)}{B(\theta, N^\star)}$$

$$\leq 2\left(\frac{\theta}{\theta^\star}\right)^d \leq 2\left(\frac{\theta_0}{\theta^\star}\right)^d.$$

$\square$

# F  Unknown RKHS norm

Within this section, we show how our algorithm can be extended to handle the case of an unknown RKHS norm. Let us define the following candidate-suggesting function for the RKHS norm hyperparameter.

**Definition F.1.** Lets consider the following candidate-suggesting function $v(\cdot) : \mathbb{N} \to \mathbb{R}^+$ to be a mapping for each $i \in \mathbb{N}$ of form:

$$v(i) = N_0 e^i.$$

For RKHS being selected by the candidate-suggesting function of Definition F.1 and length scale being selected by the one of Definition 3.1, we get:

**Lemma F.1.** *In the case of both RBF and $\nu$-Matérn kernel, we have that:*

$$\frac{R^{(\hat{\theta}, B(\hat{\theta}, \hat{N}))}(T)}{R^{(\theta^\star, B(\theta^\star, N^\star))}(T)} = \mathcal{O}(1).$$

*Proof.* **Case 1:** $\sqrt{\gamma_T^{\hat{\theta}}} > B(\hat{\theta}, \hat{N})$

$$\frac{R^{(\hat{\theta}, B(\hat{\theta}, \hat{N}))}(T)}{R^{(\theta^\star, B(\theta^\star, N^\star))}(T)} = \frac{\sqrt{T\gamma_T^{\hat{\theta}}} \left( \sqrt{\gamma_T^{\hat{\theta}}} + B(\hat{\theta}, \hat{N}) \right)}{\sqrt{T\gamma_T^{\theta^\star}} \left( \sqrt{\gamma_T^{\theta^\star}} + B(\theta^\star, N^\star) \right)} \leq 2 \left( \frac{\theta^\star}{\hat{\theta}} \right)^d \leq 2 \left( \frac{q(i^\star)}{q(i^\star + 1)} \right)^d = 2 \left( e^{-1/d} \right)^d = \mathcal{O}(1),$$

where $i^\star = \max\{i \in \mathbb{N} \mid q(i) \geq \theta^\star\}$ and as such we have $q(i^\star + 1) = \hat{\theta} \leq \theta^\star \leq q(i^\star)r$.

**Case 2:** $\sqrt{\gamma_T^{\hat{\theta}}} \leq B(\hat{\theta}, \hat{N})$

$$\frac{R^{(\hat{\theta}, B(\hat{\theta}, \hat{N}))}(T)}{R^{(\theta^\star, B(\theta^\star, N^\star))}(T)} = \frac{\sqrt{T\gamma_T^{\hat{\theta}}} \left( \sqrt{\gamma_T^{\hat{\theta}}} + B(\hat{\theta}, \hat{N}) \right)}{\sqrt{T\gamma_T^{\theta^\star}} \left( \sqrt{\gamma_T^{\theta^\star}} + B(\theta^\star, N^\star) \right)} \leq \frac{2\sqrt{\gamma_T^{\hat{\theta}}} B(\hat{\theta}, \hat{N})}{\sqrt{\gamma_T^{\theta^\star}} B(\theta^\star, N^\star)} \leq 2 \left( \frac{q(i^\star)}{q(i^\star + 1)} \right)^d \frac{v(j^\star + 1)}{v(j^\star)}$$

$$\leq 2 \left( e^{-1/d} \right)^d e = 2 = \mathcal{O}(1)$$

where $i^\star = \max\{i \in \mathbb{N} \mid q(i) \geq \theta^\star\}$ and $j^\star = \max\{j \in \mathbb{N} \mid v(j) \leq N^\star\}$ as such we have $q(i^\star + 1) = \hat{\theta} \leq \theta^\star \leq q(i^\star)$ and $v(j^\star + 1) = \hat{N} \geq N^\star \geq v(j^\star)$. $\qquad\square$

We can now prove the regret bound.

**Theorem F.1.** *Let us use confidence parameters of $\xi_t = 2\sigma_N^2 \log(d \ln g(T) \log b(T)\pi^2 t^2) - \log(3\delta)$ and $\beta_t^{\theta, N} = B(\theta, N) + \sigma_N \sqrt{2(\gamma_{t-1}^\theta + 1 + \ln(2/\delta))}$, then Algorithm 3 achieves with probability at least $1 - \delta$ the cumulative regret $R_T$ of the algorithm admits the following bound:*

$$R_T = \mathcal{O}\left( (t_0 + \iota) B^\star + \right.$$

$$\left. \left( R^{(\theta^\star, B(\theta^\star, N^\star))}(T) + \sqrt{T\xi_T} \right) \left( \left( \frac{\theta_0}{\theta^\star} \right)^d \frac{N^\star}{N_0} d \ln \frac{\theta_0}{\theta^\star} \ln \frac{N^\star}{N_0} + \iota \right) \right),$$

*where $t_0 = \max\{g^{-1}\left( e^{-1/d}\theta_0/\theta^\star \right), b^{-1}(N^\star/N_0)\}$ and $\iota = d \ln g(T) \log b(T)$.*

*Proof.* Similarly as in the proof of Theorem 4.1, we look for $t_0$, such that at least one well-specified hyperparameter value will enter the considered set. This happens after at most $t_0 = \max\{g^{-1}(\frac{\theta_0}{\theta^\star e^{1/d}}), b^{-1}(\frac{B}{B_0}e)\}$. Observe that the set of all hyperparameters introduced by time step $T$ is $A = \Theta_T \times \mathcal{B}_t$. We thus have:

$$R_T \leq 2t_0 B + |A|B + \sum_{t \notin \mathcal{T}} r_t.$$

---

**Algorithm 3** Length scale and Norm Balancing GP-UCB (LNB-GP-UCB)

---

**Require:** suspected regret bounds $R^u(\cdot)$;
    length scale growth function $g(\cdot)$; norm growth function $b(\cdot)$;
    length scale candidate-proposing function $q(\cdot)$;
    norm candidate-proposing function $v(\cdot)$;
    initial length scale $\theta_0$; initial norm $B_0$;
    confidence parameters $\{\xi_t\}_{t=1}^T$ and $\{\beta_t^u\}_{t=1}^T$
1: Set $\mathcal{D}_0 = \emptyset$, $U_1 = \{(\theta_0, B_0)\}$, $S_0^u = \emptyset$ for all $u \in U_1$
2: Set $\Theta_1 = \{\theta_0\}$, $\mathcal{B}_1 = \{B_0\}$
3: **for** $t = 1, \ldots, T$ **do**
4:    Select hyperparameter $u_t = \arg\min_{u \in U_t} R^u(|S_{t-1}^u| + 1)$
5:    Select point to query $\boldsymbol{x}_t = \arg\max_{\boldsymbol{x} \in \mathcal{X}} \mathrm{UCB}_{t-1}^{u_t}(\boldsymbol{x})$
6:    Query the black-box $y_t = f(\boldsymbol{x}_t)$
7:    Update data buffer $\mathcal{D}_t = \mathcal{D}_{t-1} \cup (x_t, y_t)$
8:    For each $u \in U_t$, set $S_t^u = \{\tau = 1, \ldots, t : u_\tau = u\}$
9:    Initialise hyperparameter sets for new iteration $U_{t+1} := U_t$, $\Theta_{t+1} := \Theta_t$, $\mathcal{B}_{t+1} := \mathcal{B}_t$
10:    **if** $\forall_{u \in U_t} |S_t^u| \neq 0$ **then**
11:        Define $L_t(u) = \left( \frac{1}{|S_t^u|} \sum_{\tau \in S_t^u} y_\tau - \sqrt{\frac{\xi_t}{|S_t^u|}} \right)$
12:        $U_{t+1} = \left\{ u \in U_t : L_t(u) + \frac{2}{|S_t^u|} \sum_{\tau \in S_t^u} \beta_\tau^u \sigma_{\tau-1}^u(\boldsymbol{x}_\tau) \geq \max_{u' \in U_t} L_t(u') \right\}$
13:    **end if**
14:    **if** $q(|\Theta_t| + 1) < \frac{\theta_0}{g(t)}$ **then**
15:        $\Theta_{t+1} = \Theta_{t+1} \cup q(|\Theta_t| + 1)$
16:        $U_{t+1} = U_{t+1} \cup (q(|\Theta_t| + 1) \times \mathcal{B}_{t+1})$
17:    **end if**
18:    **if** $v(|\mathcal{B}_t| + 1) < B_0 b(t)$ **then**
19:        $\mathcal{B}_{t+1} = \mathcal{B}_{t+1} \cup v(|\mathcal{B}_t| + 1)$
20:        $U_{t+1} = U_{t+1} \cup (v(|\mathcal{B}_{t+1}| + 1) \times \Theta_{t+1})$
21:    **end if**
22: **end for**

---

We now apply Lemma B.5 to get:

$$\sum_{t \notin \mathcal{T}} r_t \leq \mathcal{O}\left( |A|B + \left( R^{u^*}(T) + \sqrt{T\xi_T} \right) \left( \sum_{u \in \mathcal{M}_0} \sqrt{\frac{|S_T^u|}{|S_T^{u^*}|}} + |A| \right) \right),$$

where now $|A| = |\Theta_T||\mathcal{B}_T| = q^{-1}(\frac{\theta_0}{g(T)})v^{-1}(N_0 b(T)) = d\log(g(T))\log(b(T))$. We now derive a Lemma similar to Lemma D.1.

**Lemma F.2.** *If the event of Lemma B.5 holds, then for any $\theta \in \mathcal{M}_0$ and $t \geq t_0$ we have that*

$$\sqrt{\frac{|S_t^u|}{|S_t^{u^*}|}} \leq \left( \frac{\theta_0}{\theta^\star} \right)^d \frac{N^\star}{N_0}.$$

Plugging expression for $|A|$, using Lemmas F.1 and F.2 and the fact that $|\mathcal{M}_0| = \mathcal{O}(d\ln\theta^\star \ln N^\star)$ finishes the proof.

$\square$

# G   Proof of Lemma F.2

**Lemma F.2.** *If the event of Lemma B.5 holds, then for any $\theta \in \mathcal{M}_0$ and $t \geq t_0$ we have that*

$$\sqrt{\frac{|S_t^u|}{|S_t^{u^*}|}} \leq \left( \frac{\theta_0}{\theta^\star} \right)^d \frac{N^\star}{N_0}.$$

*Proof.* If $|S_t^{u^*}| \geq |S_t^u|$, the bound holds trivially. Thus we will assume $|S_t^{u^*}| < |S_t^u|$. The suspected regret bounds are of the form:

$$R^u(t) = \sqrt{T\gamma_T^\theta}\left(\sqrt{\gamma_T^\theta} + B(\theta, N)\right).$$

Due to the regret balancing condition (Lemma 5.2 of [33]), we must have:

$$R^u(|S_t^u|) \leq 2R^{u^*}(|S_t^{u^*}|)$$

$$\sqrt{\frac{|S_t^u|}{|S_t^{u^*}|}} \leq 2\frac{\sqrt{\gamma_{|S_t^{u^*}|}^{\theta^\star}}\left(\sqrt{\gamma_{|S_t^{u^*}|}^{\theta^\star}} + B(\theta^\star, N^\star)\right)}{\sqrt{\gamma_{|S_t^u|}^\theta}\left(\sqrt{\gamma_{|S_t^u|}^\theta} + B(\theta, N)\right)}$$

**Case 1:** Consider the case when $\sqrt{\gamma_{|S_t^{\theta^\star}|}^{\theta^\star}} \geq B(\theta^\star, N^\star)$. We then have:

$$\sqrt{\frac{|S_t^u|}{|S_t^{u^*}|}} \leq 2\frac{\gamma_{|S_t^{u^*}|}^{\theta^\star}}{\gamma_{|S_t^u|}^\theta} \leq 2\left(\frac{\theta}{\theta^\star}\right)^d\frac{\gamma_{|S_t^u|}^\theta}{\gamma_{|S_t^u|}^\theta} = 2\left(\frac{\theta}{\theta^\star}\right)^d \leq 2\left(\frac{\theta_0}{\theta^\star}\right)^d\frac{N^\star}{N_0}.$$

**Case 2** Consider the case when $\sqrt{\gamma_{|S_t^{u^*}|}^{\theta^\star}} < B(\theta^\star, N^\star)$. We then have:

$$\sqrt{\frac{|S_t^u|}{|S_t^{u^*}|}} \leq 2\frac{\sqrt{\gamma_{|S_t^{u^*}|}^{\theta^\star}}B(\theta^\star, N^\star)}{\sqrt{\gamma_{|S_t^u|}^\theta}B(\theta, N)} \leq 2\left(\frac{\theta}{\theta^\star}\right)^{d/2}\sqrt{\frac{\gamma_{|S_t^u|}^{\theta^\star}}{\gamma_{|S_t^u|}^\theta}}\frac{B(\theta^\star, N^\star)}{B(\theta, N)} = 2\left(\frac{\theta}{\theta^\star}\right)^d\frac{N^\star}{N}$$

$$\leq 2\left(\frac{\theta_0}{\theta^\star}\right)^d\frac{N^\star}{N_0}.$$

$\square$

# H  Derivation of optimality rates

To obtain rates for A-GP-UCB, we use Corrolary 3 of [6]. While A-GP-UCB considered the case of unknown norm and bound simultaneously, to obtain the rate for unknown length scale only, we ignore the growth function used for the norm. Note that since, for A-GP-UCB $R_T = \mathcal{O}(b(T)g(T)^d R^{u^*}(T))$ and in BO $R^u(T) = \sqrt{T\gamma_T^u}(\sqrt{B^u} + \sqrt{\gamma_T^u})$, if $b(T)g(T)^d$ grows at least as fast as $\sqrt{T}B$, then bound on $R_T$ grows at least as fast as $B^{u^*}T$ and becomes trivial. Thus for the regret bound of A-GP-UCB to be meaningful, we have to assume $b(T)g(T)^d$ grows slower than $\sqrt{T}B$.

Inspecting the bounds of LB-GP-UCB and LNB-GP-UCB in Theorems 4.1 and F.1, we see that the term with $R^{\theta^\star}(T)$ or $R^{(\theta^\star, B(\theta^\star, N^\star))}(T)$ will dominate the bound. This is because by the previous assumption on the growth of $b(T)g(T)^d$, we get that $\iota = \mathcal{O}(\ln b(T)d\ln g(T)) \leq \mathcal{O}(\ln(BT))$ and $\sqrt{T\xi_t} = \mathcal{O}(\sqrt{T\log\ln b(T)d\ln g(T)}) = \mathcal{O}(\sqrt{T\log\log TB})$ and in both RBF and $\nu$-Matérn cases regret bound grows at least as fast as $\sqrt{T\log T}$. Also the term $\left(\frac{\theta_0}{\theta^\star}\right)^d\frac{N^\star}{N_0}d\ln\frac{\theta_0}{\theta^\star}\ln\frac{N^\star}{N_0}$ is a constant and will eventually get dominated by $\iota$. We thus get that the bound will become dominated by $\iota R^{\theta^\star}(T)$ or $\iota R^{(\theta^\star, B(\theta^\star, N^\star))}(T)$ and the suboptimality is just $\iota$.

# I   Experiments Details

We used the code of [20] for computations of maximum information gain.

## I.1   Compute Resources

To run all experiments we used a machine with AMD Ryzen Threadripper 3990X 64-Core Processor and 252 GB of RAM. No GPU was needed to run the experiments. We were running multiple runs in parallel. To complete one run of each method we allocated four CPU cores. Individual runs lasted up to seven minutes for each of the methods, except for MCMC runs, which could last up to an hour (see Table 2 below).

## I.2   Running times

Table 2: Comparison of running types of different methods on each test function/ benchmark. Values after $\pm$ are standard errors over seeds.

| Function/ Benchmark | Method | Running Time (seconds) |
|---|---|---|
| Berkenkamp Function | MLE | $438 \pm 0.66$ |
| | A-GP-UCB | $443 \pm 1.51$ |
| | LB-GP-UCB | $442 \pm 1.68$ |
| | MCMC | $1653 \pm 25.99$ |
| Michalewicz Function | MLE | $237 \pm 2.41$ |
| | A-GP-UCB | $167 \pm 0.88$ |
| | LB-GP-UCB | $181 \pm 0.47$ |
| | MCMC | $3388 \pm 369.38$ |
| Crossed Barrel Materials Experiment | MLE | $55 \pm 0.10$ |
| | A-GP-UCB | $48 \pm 0.40$ |
| | LB-GP-UCB | $48 \pm 0.50$ |
| | MCMC | $471 \pm 25.20$ |
| AGNP Materials Experiment | MLE | $53 \pm 0.05$ |
| | A-GP-UCB | $49 \pm 0.18$ |
| | LB-GP-UCB | $49 \pm 0.16$ |
| | MCMC | $246 \pm 3.56$ |

