# OpenReview forum: "Bayesian Optimisation with Unknown Hyperparameters: Regret Bounds Logarithmically Closer to Optimal"
_NeurIPS.cc/2024/Conference — NeurIPS 2024 poster_

### Official Review · Reviewer_YCMG · 2024-07-09

**Soundness:** 3
**Presentation:** 2
**Contribution:** 3
**Rating:** 6
**Confidence:** 3

**Summary:**

This paper considers the Bayesian optimization (BO) problem under an unknown length scale and upper bound of the RKHS norm.
The proposed algorithm LB-GP-UCB is designed to select length scale
from certain candidates set adaptively. Furthermore, the algorithm eliminates some candidate length scales if certain conditions are met. The validity of the proposed algorithm is given both empirically and theoretically. Specifically, the proposed algorithm has
a favorable property compared with the existing A-GP-UCB algorithm in the sense of regret optimality.

**Strengths:**

- The motivation is well-discussed; specifically, I agree with the issue of
the A-GP-UCB algorithm described in the Introduction.
- From my view, the comparison with A-GP-UCB in the sense of regret optimality is interesting and novel.
- Enough numerical evaluations are given, and the author also provides the anonymized codes with an easily reproducible style.

**Weaknesses:**

- In contrast to the existing literature (Berkenkamp, 2019), the analysis is limited to where the length scale parameters are the same among each coordinate. This relates to the limitation of the practical applicability, and I think that the comparison with A-GP-UCB should be evaluated by noting this limitation.

- The method how to calculate the quantity $R^{\theta}$ (Line 3), which contains MIG we only
know the dependence of $T$ and $\theta$ of its upper bound, is ambiguous.
As far as I see the proof of Lemma C.1 and the statement of Lemma 5.2 of Pacchiano et al.,
$R^{\theta}$ should be specified as the exact upper bound of regret; only the knowledge
of the order of the regret upper bound is insufficient to specify $R^{\theta}$.
If we cannot calculate the exact value or upper bound of $R^{\theta}$, this part can become a limitation,
and the author at least should add the discussion. As far as I see the anonymized code, the numerical experiments are done by
approximating the upper bound of MIG with the numerically optimized version of MIG (Hong et al, 2023).
Even if we follow Hong et al, 2023, I believe that the exact version of the upper bound of MIG can be computed only when the input space $\mathcal{X}$ is finite and small.

- To my understanding, the Ziomek's paper is closely related to your setting, and it seemed
that some parts of your paper (e.g., Lemma A.2) borrow their ideas; however, the relation
and comparison with their paper are not described in Related Works.

Ziomek, Juliusz, Masaki Adachi, and Michael A. Osborne. "Beyond Lengthscales: No-regret Bayesian Optimisation With Unknown Hyperparameters Of Any Type." arXiv preprint arXiv:2402.01632 (2024).


[Minor]

- L108 $x^{\ast} = \max_{x \in \mathcal{X}}$ -> $x^{\ast} = \max_{x \in \mathcal{X}} f(x)$.
- L110 ... value $\theta \in \mathbb{R}$ we denote -> ... value $\theta \in \mathbb{R}$. We denote
- L117 ... them below:: -> ... them below:
- L119 $\theta \in \\{0, \theta_0\\}$ -> $\theta \in (0, \theta_0]$.
- L129 $\gamma(k^{\theta})\_{t-1}$ -> $\gamma_{t-1}(k^{\theta})$
- L129 $\frac{1}{2} \ln |I + \sigma_N^2 K_T^{\theta^{\ast}}|$ -> $\frac{1}{2} \ln |I + \sigma_N^{-2} K_T^{\theta}|$.
- Footnote 1 and Lemma 4: This statement does not hold for any stationary kernel (see Assumption 2 in Bull [11]).
I recommend the author explicitly assume RBF or Mat\'ern kernel in these statements.
- Liu et al., 2023 related to your setting. Although the unknown hyperparameter they consider is different from the setting
of your paper, I recommend adding their paper as the related work.
    - Liu, Yusha, and Aarti Singh. "Adaptation to Misspecified Kernel Regularity in Kernelised Bandits." International Conference on Artificial Intelligence and Statistics. PMLR, 2023.

**Questions:**

- Is there a possibility of extending the algorithm to a case where length scale parameters are different among each coordinate?

- In Proposition 2.3, can the author update the Berkenkamp's MIG upper bound of Mat\'ern to $\tilde{O}(T^{\frac{d}{2\nu + d}})$?
The recent result (Vakili et al., 2021) shows that the maximum information gain of Mat\'ern kernel increases
with $\tilde{O}(T^{\frac{d}{2\nu + d}})$. Their results do not provide explicit dependence
of $\theta$; however, by combining Theorem 3 of Vakili et al. and the eigendecay rate of the kernel with explicit dependence of $\theta$,
I guess that $\gamma_T = \tilde{O}(\theta^{-2\nu + d} T^{\frac{d}{2\nu + d}})$ can be obtained.
    - Vakili, Sattar, Kia Khezeli, and Victor Picheny. "On information gain and regret bounds in gaussian process bandits." International Conference on Artificial Intelligence and Statistics. PMLR, 2021.

- The exact upper bound of the MIG is obtained by relying on the uncertainty sampling as described in Section 5.1 of Srinivas's paper.
Does any problem occur with using Srinivas's results in the author's analysis?
    - Srinivas, Niranjan, et al. "Gaussian process optimization in the bandit setting: No regret and experimental design." arXiv preprint arXiv:0912.3995 (2009).

**Limitations:**

- The analysis is limited to the case where the length scale parameters are the same among each coordinate.
- No potential negative societal impact is seen.

---

> ### Author Rebuttal · Authors · 2024-08-07
>
> We would like to thank the reviewer for reading our paper and for mentioning relevant related work. We address each question/concern below.
>
> **Case of different length scales**
>
> The method of Berkenkamp (2019) cannot really handle kernels with differing length scales across coordinates. While technically A-GP-UCB can be applied to non-isotropic kernels, it decreases the lengthscale value for each coordinate at the same pace (e.g. see eq. 9 in (Berkenkamp, 2019)). As such, unless the initial length scales $\theta_0$ differ for each coordinate (which would imply strong some prior information on the relative importance of coordinates), A-GP-UCB utilises the same length scale value for each coordinate at each timestep, making it no better than an algorithm utilising the isotropic kernel. As such, our algorithm is not inferior to A-GP-UCB on that regard.
>
> As we mention in Section 7, to the best of our knowledge, there have been no results in the literature deriving the MIG bounds for the non-isotropic kernels. If one were to obtain such bounds, we believe we could very easily extend our algorithm LB-GP-UCB to handle multiple unknown length scales, in the same way as we extended it to the case of simultaneously handling unknown norm and length scale. Such an algorithm would truly handle the non-isotropic case and would be able to potentially choose to fit a GP model with different length scale values, unlike A-GP-UCB.
>
> **Updating MIG bound**
>
> We would like to thank the reviewer for mentioning that the improved MIG bounds exist, as we were not aware of that fact. **We will replace the MIG bounds with the improved ones in our Theorems.**
>
> **Relation to Ziomek et al, 2024**
>
> The problem setting considered in (Ziomek et al, 2024) differs from ours, as they assumed there are a number of candidate hyperparameter values given to us at the start of the problem, making the problem much easier. Their proposed algorithm also scales with the MIG of the worst candidate and as such could be arbitrarly far from the optimal. **We will add this discussion to the Related Work section.**
>
> **Knowledge of Regret Bounds**
>
> One do not need the exact form of the regret bounds---the knowledge up to a constant is sufficient. The fact that the algorithm selects hyperparameters by the rule in line 3 is only used in two places in the proof of the Theorem 4.1 The first place is Lemma C.1., where the constants $C$ in the regret bounds cancel out. The second place is in line 472 - 474 on page 14, where we assume that for the regret bounds we must have $R(t+1) < R(t) + 2B$, since $2B$ is the highest instantaneous regret we can possibly suffer. If the order dependence is known, one can always find such a constant $C$ such that this constraint is respected.
>
> While the lack of need for exact knowledge of $C$ might seem surprising at first glance, it is a direct result of the fact the regret bound for LB-GP-UCB is also derived up to a constant factor. Changing $C$ in the suspected regret bound does not affect the order dependence of the regret bound of LB-GP-UCB. This is a very interesting point and we are grateful to the reviewer for mentioning it. **We will add a short discussion in the paper to clarify this.**
>
> **Applicability of the MIG bounds from (Srinivas et al, 2009)**
>
> The results from Srinivas et al (2009) rely on uncertainty sampling to derive a bound on the sum of predictive variances for **any** strategy (the MIG bound). In fact, Srinivas et al (2009) use this MIG bound to derive a regret bound for the GP-UCB algorithm. If each of the GP-UCB base algorithms were run in isolation, these bounds clearly hold. These bounds must therefore also hold in the case, where datapoints are shared between algorithms, as variance is a non-increasing function of the number of conditioning datapoints.
>
> **Minor points**
>
> We would like to thank the reviewer for pointing out the typos and that Lemma 4 of [11] requires an additional assumption. **We will correct the typos and explicitly restrict the statement to RBF or Matern kernel.** We would also like to thank for pointing out more relevant, related work, **we will add it to the related work section**.

---

> > ### Comment · Reviewer_YCMG · 2024-08-09
> >
> > I thank the reviewer for responding to my questions and correcting some of my misunderstandings. I have raised my score to 6.

---

> > > ### Author Response · Authors · 2024-08-11
> > >
> > > Thank you very much for taking the time to review our paper and respond to us. We are glad to know you are satisfied with our rebuttal.

---

### Official Review · Reviewer_k1W2 · 2024-07-10

**Soundness:** 3
**Presentation:** 3
**Contribution:** 4
**Rating:** 7
**Confidence:** 4

**Summary:**

This paper proposes a novel Bayesian optimization algorithm for the setting with unknown kernel lengthscale. The proposed approach improves upon prior work by running a logarithmic array of algorithms on exponentially decreasing lengthscales in combination with a regret balancing scheme. The paper proves a regret guarantee with an exponential improvement compared to prior work in the ratio of the regret compared to the oracle algorithm that knows the optimal lengthscale. The experimental evaluation also shows substantial improvements compared to prior the prior work.

**Strengths:**

Originality: This paper introduces a clever regret balancing scheme to obtain a improved regret bound for Bayesian optimization with unknown kernel lengthscale. While the individual ideas already appear in prior work (e.g. regret balancing, and decreasing the lengthscale), the combination in this setting is novel.

Quality & Clarity: The paper is overall well written but could benefit from careful proof reading (some suggestions below). The main idea is easy to follow and there is an extensive comparison to the prior work. The regret bound is a significant improvement over prior work. The experimental evaluation shows that the proposed approach outperforms the prior work.

Significance: The findings are of high interest to the Bayesian optimization community (and users of Bayesian optimization) as selecting hyperparameter remains a major challenge. In particular, classical schemes (like marginal likelihood) tend to fail because there is too little data initially.

**Weaknesses:**

The paper could use some polishing for English and punctuation (e.g. many sentences are quite long). In addition, the authors could provide more intuition (e.g. why regret balancing works, and the elimination scheme) and try to make the paper more accessible to readers not familiar with the tools used.

 Minors:
* line 110: punctuation
* Algorithm 1, line 7: Is there a simpler notation for the set $S_t^\theta$ ?
* 209: Long sentence, unclear what "their" refers to.
* 229: Wording/missing word "highest ? as possible"

**Questions:**

N/A

**Limitations:**

The main limitation is that the approach so far only works for a scalar parameter that induces a natural nesting of Hilbert spaces.

---

> ### Author Rebuttal · Authors · 2024-08-07
>
> We would like to thank the reviewer for reading our submission, and pointing out the issues with writing. **We will correct the writing errors pointed out by the reviewer and try to break long sentences into shorter ones.** We agree that the algorithm is quite complex and it would be good to give more intuition to the readers. **We plan to use the additional page in the camera-ready version to add a longer, more intuitive explanation of the regret balancing scheme with some easily-understandable figures.**
>
> When it comes to the definition of set $S_t^\theta$, in the pseudocode, it would be possible to drop the iteration subscript $t$ and just redefine the set at each iteration. However, we explicitly wanted to index this set by the current timestep $t$ and the hyperparameter value $\theta$, as this notation is very convienient later in the proofs.

---

> > ### Comment · Reviewer_k1W2 · 2024-08-12
> >
> > I'd like to thank the authors for the response and clarifications. I read the reviews and rebuttal, and my evaluation remains positive.

---

> > > ### Author Response · Authors · 2024-08-13
> > >
> > > Thank you very much for taking the time to review our paper and respond to us. We are glad to know you are satisfied with our clarifications.

---

### Official Review · Reviewer_NyJf · 2024-07-11

**Soundness:** 2
**Presentation:** 1
**Contribution:** 3
**Rating:** 6
**Confidence:** 3

**Summary:**

This paper introduces LB-GP-UCB (Lengthscale Balancing GP-UCB), a Bayesian Optimization (BO) algorithm that proposes a new tuning of the covariance function hyperparameters. A regret bound is derived, with logarithmic improvement over A-GP-UCB, the most similar solution in the literature. Some numerical experiments also show improvements in practice.

**Strengths:**

LB-GP-UCB is a significant improvement over A-GP-UCB, and constitutes an interesting step forward in the field of no-regret BO with unknown hyperparameters.

**Weaknesses:**

My concerns are mostly related to the experimental part of the paper.

**Missing Baselines**. In Section 6, the authors seem to discard many solutions addressing the problem of BO with unknown hyperparameters because they do not provide any theoretical analysis for their algorithms (see [13, 20, 25, 28] in the main paper). Although I understand that LB-GP-UCB comes with an additional, reassuring theoretical guarantee, its empirical performance should still be compared against some empirical algorithms at least.

**Missing Benchmarks**. Only four benchmarks were considered, I think that is not enough for a comprehensive study of LB-GP-UCB's empirical behavior. I believe additional experiments should be run.

**Impact of the Dimension $d$**. The dimensionality of the objective function may have an important impact on the performance of LB-GP-UCB. However, the dimensionality of the problems was not specified in the main text nor in the Appendix E entitled "Experimental Details".

**Wall-Clock Time Comparison**. In Appendix E, it is mentionned that all experiments for every method (except MCMC) took up to 4 minutes to run, but I would be interested in the precise wall-clock time for each solution and each experiment. This is important as online estimation of the hyperparameters brings a computational overhead to the BO algorithm.

**Questions:**

Here are some questions to spark the discussion with the authors.

(1) Have you compared LB-GP-UCB to any of the algorithms presented in [13, 20, 25, 28]? If not, on what ground have you discarded them for your empirical evaluation?

(2) Why have you chosen these 4 benchmarks? I know that the rebuttal period is very short, but I believe more experimental results on a variety of problems (e.g., different smoothness of the objective function, different dimensionality...) should be considered to strengthen Section 5.

(3) What were the dimensionality for the considered benchmarks? Do you have any insight on how LB-GP-UCB would react to higher-dimensional problems?

(4) Do you have the precise wall-clock times for each solution and each experiment?

**Limitations:**

I believe the authors have properly addressed the limitations of their work.

---

> ### Author Rebuttal · Authors · 2024-08-07
>
> We would like to thank the reviewer for reading our submission and providing feedback on our paper. We address each question/concern below:
>
> (1) The methods used by [13] and [28] are equivalent to the MCMC baseline we compared against, where the hyperparameters are marginalised from the acqusition function using Monte Carlo samples (as we mention at the beginning of Section 5). [25] is an empirical study of robustness to misspecification of the prior on hyperparameters that does not provide any novel method of hyperparameter estimation. The codebase of [20] has been taken down from the web, and currently no publicly available implementation of their algorithm exists.
>
> (2) We chose benchmarks with differing smoothness throughout the domain (Berkenkamp, Michalewicz) or those that exhibit "needle in a haystack" behaviour (such as Material Design problems), as typically used methods such as maximum likelihood tend to struggle on these kinds of problems. Our benchmarks were chosen to showcase how our method can be used to tackle problems with which existing methods struggle. Of course, if one were to chose lesschallenging benchmarks with constant smoothness throughout the domain, it is entirely possible the improvement delivered by LB-GP-UCB over MLE and other baseline would be smaller. We do not claim that our method would provide universal improvement across all sorts of benchmarks, but rather that MLE and MCMC can perform poorly on a certain class of problems on which our algorithm performs well. We will add this comment to the limitation section.
>
> We would also like to emphasise that the paper proposing A-GP-UCB (published in JMLR, 2019) included only two empirical benchmarks, so we would argue that our experimental evaluation of four benchmarks meets the standard for papers, whose main contribution is theoretical. In fact, Reviewer YCMG mentions "Enough numerical evaluations are given (...)".
>
>
> (3) The Berkenkamp function is a 1-dimensional synthetic function. We used a 5-dimensional version of the Michalewicz function. AGNP and CrossedBarrel are 5-dimensional and 4-dimensional real-world problems, respectively. We will amend the manuscript to include the dimensionality of each of the problems.
>
> As we prove in the paper, LB-GP-UCB can recover performance "close" to the performance of GP-UCB optimiser with the oracle knowledge of the optimal length scale value. However, it is well-known that even the regret bound of this oracle optimiser would grow with the dimensionality of the problem and same is true for the LB-GP-UCB. To remedy that, one could enhance LB-GP-UCB optimiser in the same way as standard BO optimiser is being enhanced to perform well in high-dimensional spaces (e.g. by adding a Trust Region or decomposing the input space). These enhancements are orthogonal to our method.
>
> (4) Yes, we do have wallclock times for our experiments. We provide them below and **will update the script to include the exact wallclock times in the Appendix**. All the times below are seconds, values after +/- are standard errors from accross the seeds.
>
> **Berkenkamp Function:**
>
> MLE : 438 +/- 0.66
>
> A-GP-UCB : 443 +/- 1.51
>
> LB-GP-UCB : 442 +/- 1.68
>
> MCMC : 1653 +/- 25.99
>
> **Michalewicz Function:**
>
> MLE : 237 +/- 2.41
>
> A-GP-UCB : 167 +/- 0.88
>
> LB-GP-UCB : 181 +/- 0.47
>
> MCMC : 3388 +/- 369.38
>
> **Crossed Barrel Materials Experiment:**
>
> MLE : 55 +/- 0.1
>
> A-GP-UCB : 48 +/- 0.4
>
> LB-GP-UCB : 48 +/- 0.5
>
> MCMC : 471 +/- 25.2
>
> **AGNP Materials Experiment:**
>
> MLE : 53 +/- 0.05
>
> A-GP-UCB : 49 +/- 0.18
>
> LB-GP-UCB : 49 +/- 0.16
>
> MCMC : 246 +/- 3.56
>
> As such, LB-GP-UCB is faster than MLE on all benchmarks except for Berkenkamp Function, where it is 4 seconds ($\approx 1$ \%)  slower. MCMC is significantly slower than any other baseline.

---

> > ### Comment · Reviewer_NyJf · 2024-08-09
> > **Rebuttal Ack**
> >
> > Thank you for the clarifications.
> >
> > I am now positive about the paper and I have increased my score to 6.

---

> > > ### Author Response · Authors · 2024-08-11
> > >
> > > Thank you for taking the time to review the paper and to respond to us. We are glad to know that our rebuttal addressed your concerns.

---

### Official Review · Reviewer_5KYZ · 2024-07-19

**Soundness:** 3
**Presentation:** 3
**Contribution:** 3
**Rating:** 7
**Confidence:** 3

**Summary:**

This paper proposes an approach to deal with unknown hyper-parameters in Gaussian process upper confidence bound (GP-UCB) algorithms, a popular Bayesian optimisation (BO) strategy. The objective function is assumed to be a member of a reproducing kernel Hilbert space (RKHS) associated with a translation-invariant kernel class whose length-scale is unknown. Algorithms are proposed to adaptively estimate the unknown kernel length-scales and an upper bound on the RKHS norm of the objective function, which are common hyper-parameters for GP-UCB algorithms. Theoretical guarantees on the regret for the proposed algorithms are provided, which show that rate of the algorithm's cumulative regret to the regret of an algorithm with knowledge about the exact hyper-parameters are only logarithmic in contrast to previous approaches. Experimental results are presented comparing the regret of the proposed algorithms to typical hyper-parameter estimation strategies in the BO literature.

**Strengths:**

* The paper builds well on existing theoretical results and a novel rigorous analysis.
* Experimental results show improvements against existing popular hyper-parameter estimation strategies, bringing new insights.
* Existing relevant literature seems well covered by related work section.
* The text is well organised following a mostly clear structure.

**Weaknesses:**

* It is unclear how close the estimated length-scale and RKHS norm are to their true values at each iteration.
* There are no (theoretical) convergence results on the algorithm’s estimates, only the regret bounds.
* The proposed algorithms are only compared against other GP-UCB strategies. It’d be interesting to see how they compare to other methods which do not require explicitly knowledge of the RKHS norm bound, for example, such as expected improvement algorithms. There are also no comparisons against meta-learning BO strategies. Even though they require prior data, it’d be interesting to see how close (or better) the performance of the proposed algorithms can get to them.

**Questions:**

* Another important hyper-parameter that might affect GP-UCB algorithms' performance is the sub-Gaussian noise parameter upper bound. However, the paper presents no discussion about the noise parameter. I was wondering if the authors have considered estimation strategies for that hyper-parameter as well.

**Limitations:**

Discussions on the main limitations of the theoretical analysis are presented, but there are no discussions on scalability issues, such as problems involving high-dimensional data or large datasets, which often require low-rank GP approximations. There are also no discussions on noise hyper-parameters (i.e., the sub-Gaussian noise parameter), another important hyper-parameter that might be unknown for some applications.

---

> ### Author Rebuttal · Authors · 2024-08-07
>
> We would like to thank the reviewer for reading our submission and for their appreciation for our theoretical analysis and empirical evaluation. We address each question/concern below.
>
> **Convergence of length-scales estimates**
> In general, we consider the setting in which the function can arbitrarly change its smoothness throughout the domain. As such, it is impossible to guarantee correctness of the length scale estimation unless we cover the input space with an infinitely-dense grid---however, doing so would defeat the purpose of conducting sample-efficienct optimisation. Our algorithm side-steps the need for accurate length scale estimation by appriopriately balancing the regret. As such, we do not consider it a weakness that we do not solve the impossible task of precisely estimating length scales---instead, we propose an alternative approach that solves a practically-relevant problem.
>
> Also, as shown in Figures 2 and 4, in practice, the length scale values typically selected by our algorithm are close to the estimates of optimal length scale value. As such, while we cannot guarantee the convergance of that estimator (which, as explained before, would be impossible in the considered setting), we observe reasonable convergence empirically.
>
> **Comparison to Exptected Improvement**
> While it is true that EI does not require knowledge of the RKHS norm to compute the acqusition function, in practice, EI still requires the specification of output scale value $c$ for the kernel $c k(x,x^\prime)$. We have that $||f||_{ck} = \frac{||f||_{k}}{c}$, so if we do not know the RKHS norm, we still have one more parameter to find. Additionally, EI still requires knowledge of the length scale. Our algorithm can be used to remedy these problems also in the case of EI, where each of the base learners could be an instance of GP-EI instead of GP-UCB.
>
> **Comparison to Meta-Learning strategies**
> The only previous work of which we are aware that solves the problem of unknown hyperparameters via meta-learning is the work of , which we cite in the related work section. They assume the training and target function were sampled from the same Gaussian Process prior, which is a different setting from ours, were we do not impose any prior on the black-box function.
>
> We did not compare against meta-learning baselines, as such approaches are only applicable where one can easily find functions that are highly "similar" to the target functions. For the benchmarks we considered, it is not clear how to find such similar functions.
>
>
> **Case of unknown noise**
> The problem of simultaneously estimating noise magnitude and length scale value is ill-posed as, for example, pointed by [Karvonen and Oates, 2023]. An intuitive way to see that this problem is ill-posed is by observing that if the function values we observe change rapidly, we can never know for sure if the change is caused by the true function $f(\cdot)$ changing rapidly (implying a short length scale value) or by the magnitude of noise being large and function changing slowly (implying long length scale value). As such, simultanous estimation of noise magnitude and kernel hyperparameters is likely impossible. However, we agree that we should comment on this, when discussing limitations. **We will update the paper to reflect that.**
>
> **Scalability Limitations**
> Scalability to high-dimensional spaces and large datasets is a limitation of virtually all optimisation methods based on standard Gaussian Process models. These limitations are not particularly tied to the method we propose. **However, for clarity, we will update the script to clearly mention those issues** as a limitation of the standard Gaussian Process model and, by extension, a limitation of our algorithm, which relies on that model.
>
>
> **References**
> Karvonen, Toni, and Chris J. Oates. "Maximum likelihood estimation in Gaussian process regression is ill-posed." Journal of Machine Learning Research 24.120 (2023): 1-47.

---

> > ### Comment · Reviewer_5KYZ · 2024-08-12
> >
> > Thanks for addressing my concerns. I keep my vote for this paper to be accepted, as it brings an important contribution to the BO community.

---

> > > ### Author Response · Authors · 2024-08-13
> > >
> > > Thank you very much for taking the time to review our paper and respond to us. We are glad to know our rebuttal addressed your concerns.

---

### Author Rebuttal · Authors · 2024-08-07

We would like to thank the reviewers for taking time to read our submission and provide insightful feedback, as well as asking interesting question. We answer to each of the reviewers individually below.

---

### Decision · Program_Chairs · 2024-09-25

**Decision:**

Accept (poster)

**Comment:**

The paper introduces LB-GP-UCB, a novel Bayesian Optimization algorithm designed to address the challenge of unknown hyperparameters, particularly the kernel length scale, in Gaussian Process models. The key contribution is a regret minimization approach that shows logarithmic improvement over existing methods, such as A-GP-UCB. The reviewers agree that the theoretical advancements are significant, and the empirical results, though limited, demonstrate the method's effectiveness on selected benchmarks.

The paper could benefit from clearer mathematical presentation and discussions on practical issues like scalability. Should the paper be accepted, the authors are encouraged to expand the experimental evaluation and refine the clarity of their presentation.